# Embryonic hematopoiesis modulates the inflammatory response and larval hematopoiesis in *Drosophila*

**Wael Bazzi**[1,2,3,4†], **Pierre B Cattenoz**[1,2,3,4†], **Claude Delaporte**[1,2,3,4], **Vasanthi Dasari**[1,2,3,4], **Rosy Sakr**[1,2,3,4], **Yoshihiro Yuasa**[1,2,3,4], **Angela Giangrande**[1,2,3,4]*

[1]Institut de Génétique et de Biologie Moléculaire et Cellulaire, Illkirch, France; [2]UMR7104, Centre National de la Recherche Scientifique, Illkirch, France; [3]U1258, Institut National de la Santé et de la Recherche Médicale, Illkirch, France; [4]Université de Strasbourg, Illkirch, France

**Abstract** Recent lineage tracing analyses have significantly improved our understanding of immune system development and highlighted the importance of the different hematopoietic waves. The current challenge is to understand whether these waves interact and whether this affects the function of the immune system. Here we report a molecular pathway regulating the immune response and involving the communication between embryonic and larval hematopoietic waves in *Drosophila*. Down-regulating the transcription factor Gcm specific to embryonic hematopoiesis enhances the larval phenotypes induced by over-expressing the pro-inflammatory Jak/Stat pathway or by wasp infestation. Gcm works by modulating the transduction of the Upd cytokines to the site of larval hematopoiesis and hence the response to chronic (Jak/Stat over-expression) and acute (wasp infestation) immune challenges. Thus, homeostatic interactions control the function of the immune system in physiology and pathology. Our data also indicate that a transiently expressed developmental pathway has a long-lasting effect on the immune response.
DOI: https://doi.org/10.7554/eLife.34890.001

*For correspondence: angela@igbmc.fr

†These authors contributed equally to this work

Competing interests: The authors declare that no competing interests exist.

## Introduction

In vertebrates, immune cells are generated from hematopoietic waves occurring at distinct places and times. The first wave occurs in the yolk sac and generates primitive erythrocytes and macrophages (*Baron et al., 2012*; *Perdiguero and Geissmann, 2016*). Later waves occur at several time points from embryonic to adult stages and produce all the blood lineages of the adult organism (*Cumano and Godin, 2007*; *Orkin and Zon, 2008*). Recent advances in lineage tracing approaches have made it possible to unambiguously assess the origin of immune cells and have shown that the immune cells from distinct hematopoietic waves coexist (*Perdiguero and Geissmann, 2016*; *Ginhoux et al., 2010*; *Gomez Perdiguero et al., 2015*; *Hoeffel et al., 2015*; *Ghosh et al., 2015*). Despite these great advances, however, the precise contribution of each wave to the immune response is still poorly understood (*Ditadi et al., 2017*). Moreover, cells from the early wave could help those from later mounting an appropriate response, calling for homeostatic interactions in physiological and pathological conditions.

In *Drosophila melanogaster* and *Manduca sexta*, a first hematopoietic wave occurs in the procephalic mesoderm during the embryonic life (*Nardi et al., 2003*; *Gold and Brückner, 2014*). Pluripotent precursors called prohemocytes generate plasmatocytes and crystal cells that constitute the only immune cells until the larval stage. The crystal cells represent 5% of the hemocytes, they remain close to the proventriculus and play role in melanization; the plasmatocytes represent 95% of the

hemocytes and are professional macrophages that populate the whole animal. During the larval life, the plasmatocytes shuttle between the hemolymph (circulating hemocytes) and a subepithelium compartment they transiently attach to (resident or sessile hemocytes) (*Gold and Brückner, 2014*; *Stofanko et al., 2008*). The second wave occurs at the larval stage in the lymph gland (*Nardi et al., 2003*; *Grigorian and Hartenstein, 2013*), which starts forming during embryogenesis and keeps growing until the onset of metamorphosis, when it disintegrates to release plasmatocytes and crystal cells into the organism. Upon an acute challenge such as wasp infestation, the larva mounts a complex immune response and similar phenotypes are observed in genetic backgrounds in which pro-inflammatory pathways such as Jak/Stat are constitutively activated, a condition that could be considered as analogous to chronic inflammation (*Luo et al., 1995*; *Harrison et al., 1995*). Typically, the resident hemocytes leave the sessile compartment and enter in circulation, hemocytes proliferate and plasmatocytes transform into lamellocytes to form cell aggregates called 'melanotic tumors' that encapsulate the parasite wasp egg, the lymph gland undergoes premature histolysis and hemocyte release (*Letourneau et al., 2016*). The sequence of these events and the contribution of embryonic hematopoiesis to this immune response are not clear.

We here describe a link in the regulation of inflammation and its threshold of response between events that, in the past, have been described as independent events separated in time and location. We show that the embryonic wave is necessary to mount an adequate immune response upon signaling to the larval wave and that the only known transcription factor specific of the embryonic hemocytes, Glide/Gcm (Gcm throughout the manuscript) (*Bernardoni et al., 1997*; *Lebestky et al., 2000*; *Alfonso and Jones, 2002*; *Bataillé et al., 2005*; *Avet-Rochex et al., 2010*) is a major player in this process. Gcm induces the expression of Jak/Stat inhibitors and in so doing inhibits the production of Jak/Stat dependent pro-inflammatory cytokines that eventually control the Jak/Stat pathway in the muscle. As a consequence, the second hematopoietic wave occurring in the lymph gland is affected as well. Thus, the embryonic hematopoietic wave contributes to the inflammatory response and modulates larval hematopoiesis, demonstrating the importance of interactions between embryonic hemocytes and the lymph gland during the inflammatory response. Finally, we highlight the long-lasting effect of a transcription factor expressed early and transiently during hematopoiesis.

## Results

### Gcm inhibits melanotic tumor formation and induces the expression of jak/Stat inhibitors

Our first step was to investigate the impact of the embryonic plasmatocytes on systemic inflammation. The Jak/Stat pathway is a major transducer of the inflammatory signal upon immune challenge. Its constitutive activation due to the mutation *hopscotch*$^{Tum-l}$ (*hop*$^{Tum-l}$) (*Luo et al., 1995*) in the Jak kinase induces the formation of melanotic tumors in larvae, a strong increase in the number of hemocytes, the production of lamellocytes (*Figure 1a–e*) and precocious lymph gland histolysis (*Luo et al., 1995*; *Harrison et al., 1995*). We crossed *hop*$^{Tum-l}$ flies with flies in which we modulated the expression of the Gcm transcription factor. Gcm is expressed early and transiently in the first hematopoietic wave, from embryonic stage 5 to stage 11, but no longer in the mature plasmatocytes (*Bernardoni et al., 1997*; *Alfonso and Jones, 2002*; *Evans et al., 2003*), nor is it expressed during the larval hematopoietic wave (*Figure 1—figure supplement 1*) (*Bernardoni et al., 1997*; *Lebestky et al., 2000*; *Alfonso and Jones, 2002*; *Bataillé et al., 2005*; *Avet-Rochex et al., 2010*; *Zaidman-Rémy et al., 2012*). Homozygous *gcm* null embryos die before hatching and show a slightly reduced number of plasmatocytes (*Bataillé et al., 2005*) with concomitant increase in crystal cell number (*Bataillé et al., 2005*) (*Figure 1—figure supplement 2a–e*), however downregulating Gcm expression with the *UAS-gcmRNAi* transgenic line (crossed with the *gcmGal4* or *gcm>* line (*gcm> gcm KD*) (*Gupta et al., 2016*) (*Figure 1—figure supplement 3a,b*) induces a milder phenotype: the animals are viable, the number of hemocytes and that of crystal cells is similar to that of control animals (*Figure 1d*, *Figure 1—figure supplement 3c–h*). Interestingly, *gcm KD* larvae contain no melanotic tumors and very rare lamellocytes (*Figure 1b,d*, *Figure 1—figure supplement 4a–c'*). Accordingly, *gcm KD* hemocytes express most plasmatocyte and few lamellocyte markers (*Figure 1f*).

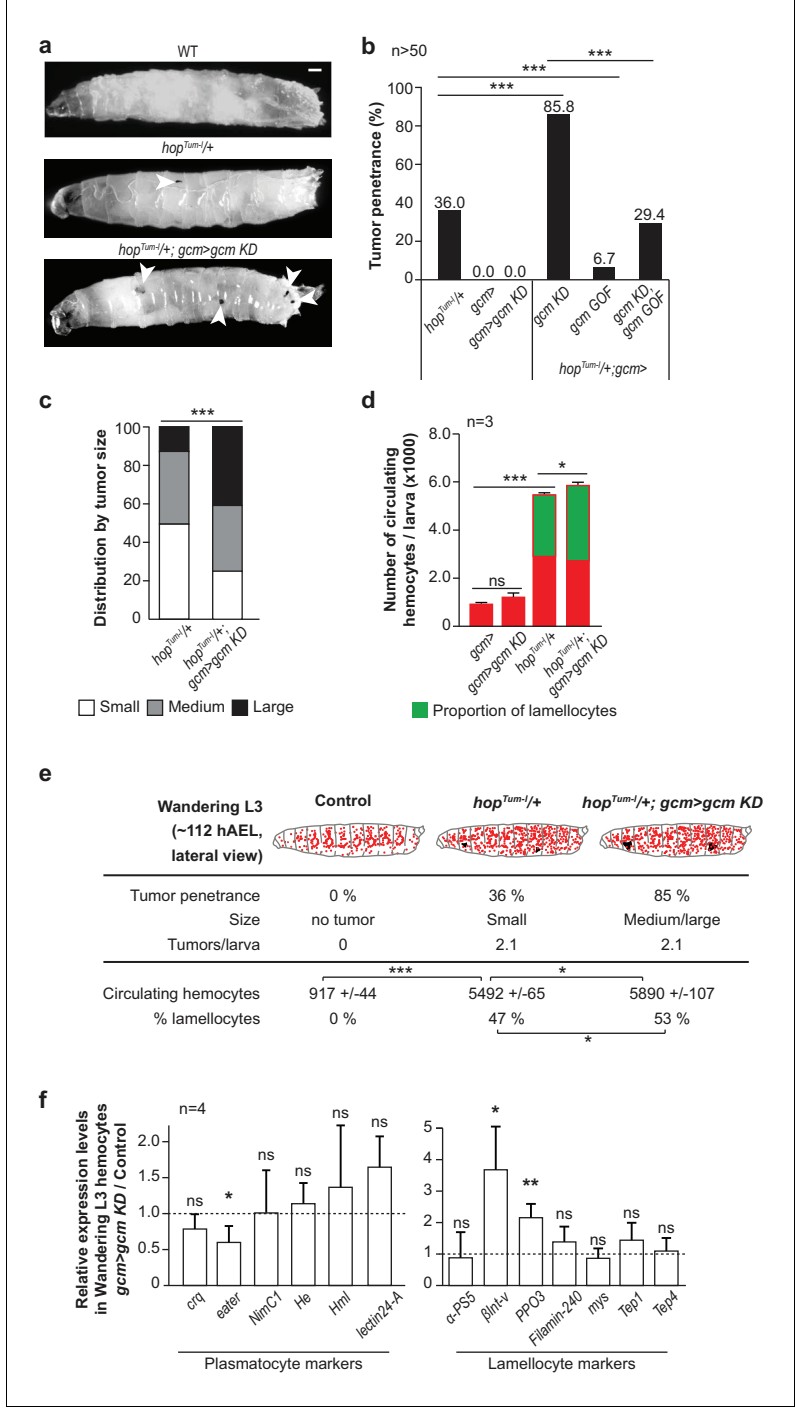

**Figure 1.** Gcm hinders Jak/Stat-mediated melanotic tumor formation. (**a**) Third instar larvae of the indicated genotypes (Note that all the detailed genotypes are in the Supplementary Methods, Fly strains and genetics section). White arrowheads indicate the melanotic tumors. (**b**) Penetrance of melanotic tumors (50). (**c**) Phenotype expressivity assessed as tumor size (n > 40). (**d**) Total number of circulating hemocytes and lamellocyte contribution (n = 3, using 10 larvae/replicate). (**e**) Hemocytes (red) and tumors (black) in third instar larvae of the mentioned genotypes. Assessment in terms of tumor penetrance, size and number of tumors/larva, number of circulating hemocytes and percentage of lamellocytes (n = 3, using 10 larvae/replicate). In all figures, *p<0.00.5, **p<0.01; ***p<0.001, ns: not significant; scale bar: 50 μm unless otherwise specified. All the P values shown in this and in the following figures are in *Figure 1—source data 1*. (**f**) Relative expression levels of plasmatocyte and lamellocyte markers in larval hemocytes *gcm> gcm KD* compared to *gcm>* , measured by qPCR. The data are

*Figure 1 continued on next page*

*Figure 1 continued*

normalized to two housekeeping genes and then the ratio (*gcm> gcm KD/gcm*) is plotted (n = 4). The error bars represent the sum of the s.e.m. of *gcm> gcm KD* and *gcm> .*

DOI: https://doi.org/10.7554/eLife.34890.002

The following source data and figure supplements are available for figure 1:

**Source data 1.** p-values for all figures.
DOI: https://doi.org/10.7554/eLife.34890.007
**Figure supplement 1.** Gcm is not expressed in the second hematopoietic wave.
DOI: https://doi.org/10.7554/eLife.34890.003
**Figure supplement 2.** Crystal cell phenotype in *gcm26* embryos.
DOI: https://doi.org/10.7554/eLife.34890.004
**Figure supplement 3.** Crystal cell phenotype in *gcm KD* larvae.
DOI: https://doi.org/10.7554/eLife.34890.005
**Figure supplement 4.** Gcm inhibits Jak/Stat-mediated melanotic tumor formation.
DOI: https://doi.org/10.7554/eLife.34890.006

Combining *gcm KD* with *hop$^{Tum-l}$* triggers much more severe defects than those shown by *hop$^{Tum-l}$* animals: the penetrance of the tumors increases from 36% to 85% (% of larvae carrying the tumors, *Figure 1b,e*), the tumors are bigger (*Figure 1c,e*, *Figure 1—figure supplement 4d*) and the larvae produce more lamellocytes (*Figure 1d,e*). In addition, animals in which the Jak/Stat pathway is constantly active and express Gcm at low levels have reduced viability, with fewer larvae hatching as compared to control animals (i.e. animals that only express one of the two molecular defects). An exacerbated tumor phenotype is also observed upon crossing *gcm KD* flies with a *gcmGal4* driver inactive in glia (*repoGal80,gcm>* or *gcm(hemo)>*), the other main territory of Gcm expression, or with independent embryonic-specific hemocyte drivers (*srp(hemo)>* and *sn>*) or upon using the *gcm26* null mutation in heterozygous conditions (*Figure 1—figure supplement 4e*). Thus, the expression of the Gcm transcription factor in the primitive hemocytes modulates the inflammatory response induced by the constitutive activation of the Jak/Stat pathway. Gcm plays an important role as its over-expression (*gcm Gain of Function* or *GOF*) rescues the *hop$^{Tum-l}$* phenotype as well as that of *hop$^{Tum-l}$;gcm KD* animals (*Figure 1b*). Interestingly, the number of circulating hemocytes is 5.5-fold higher in *hop$^{Tum-l}$* compared to wild-type animals, but the further increase observed in *hop$^{Tum-l}$;gcm KD* animals is incremental (1.1-fold), mostly due to an increase of lamellocytes' number (*Figure 1d,e*).

The hemocyte phenotype predicts that Gcm inhibits the Jak/Stat pathway. Since the activation of the pathway leads to the phosphorylation and to the nuclear translocation of the Stat transcription factor, we first transfected S2 *Drosophila* cells with a Gcm and/or with a Hop$^{Tum-l}$ expression vector and validated this prediction in vitro. Gcm represses the nuclear localization of Stat induced by *hop$^{Tum-l}$* (*Figure 2a–c'''*, *Figure 1—figure supplement 4f*). Next, we investigated the molecular mechanisms by which Gcm could affect inflammatory response. A genome-wide DamID screen identifying the direct targets of Gcm highlighted several negative regulators of that pathway (*Cattenoz et al., 2016*) and we selected three major Jak/Stat inhibitors based on their role in hematopoiesis: Ptp61F, Socs36E and Socs44A (*Müller et al., 2005*) (*Figure 2d*, *Figure 1—figure supplement 4g–i*). Ptp61F acts by de-phosphorylating Hop, the only Jak present in flies, and the transcription factor Stat92E (*Müller et al., 2005*; *Baeg et al., 2005*; *Villarino et al., 2017*). Socs36E and Socs44A belong to the suppressor of cytokine signaling family that suppresses Jak/Stat activation by competing with Stat for binding to the Jak catalytic domain (*Naka et al., 1997*; *Yasukawa et al., 1999*). Transfecting S2 *Drosophila* cells with a Gcm expression vector increases the endogenous levels of *Ptp61F*, *Socs36E* and *Socs44A* transcripts (*Figure 2e*). In addition, *gcm KD* embryos express these genes at low levels, validating Gcm as a regulator of the three genes, likely at the transcriptional level (*Figure 2f*).

Since Gcm is specifically expressed in the embryonic hemocytes, we assessed whether inhibiting the Jak/Stat pathway only in those cells affects the immune response. *hop$^{Tum-l}$* animals in which the expression of any of the three inhibitors is specifically silenced in embryonic hemocytes (*gcm>*) show a strong enhancement of the tumor penetrance (> 90%, *Figure 2g*). Accordingly, the over-expression of Ptp61F in embryonic hemocytes rescues the *hop$^{Tum-l}$* phenotype and the exacerbated phenotype observed in the *hop$^{Tum-l}$/+;gcm KD* larvae (*Figure 2g*).

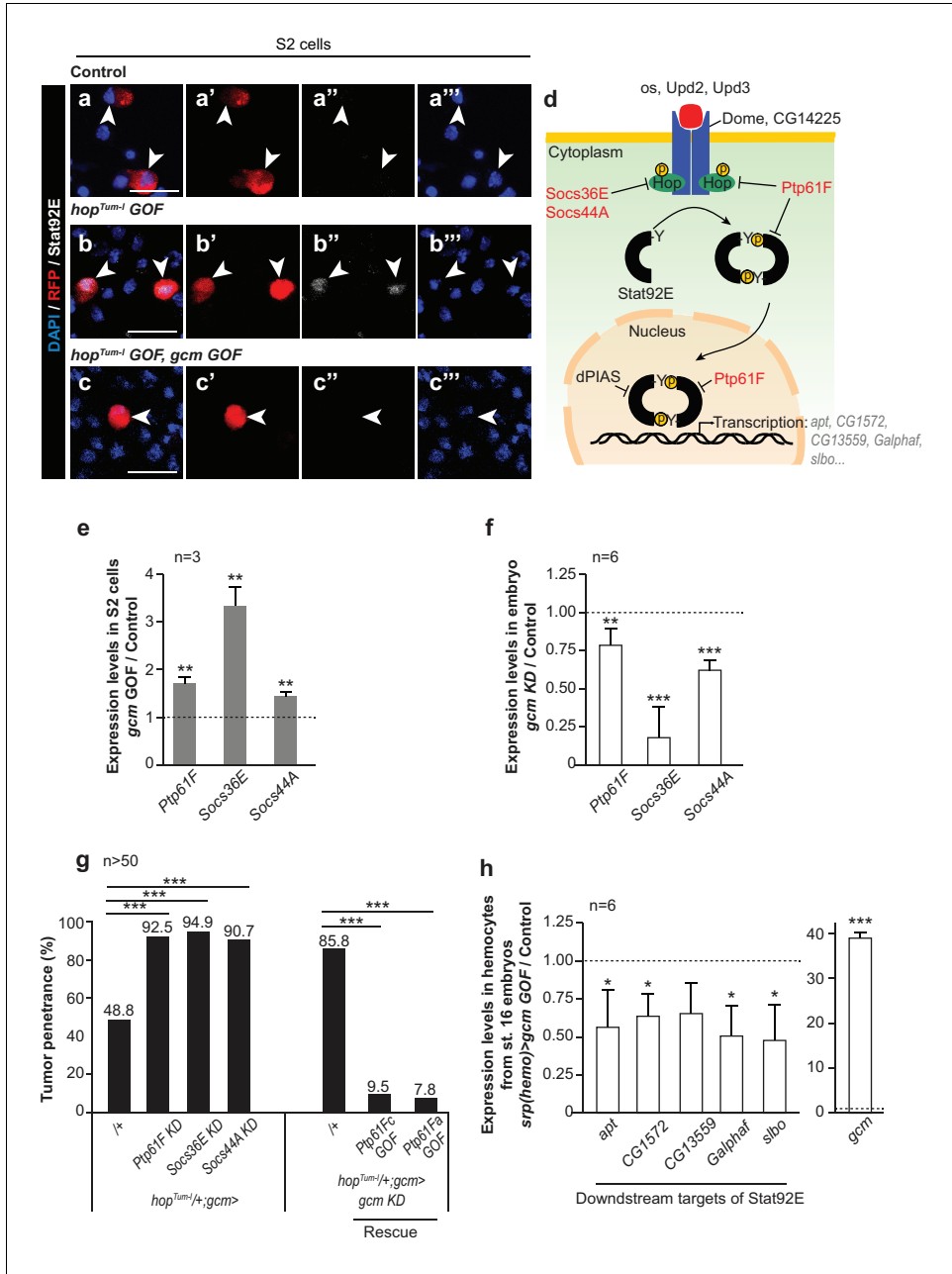

**Figure 2.** Gcm induces the expression of Jak/Stat inhibitors and hinders Jak/Stat-mediated melanotic tumor formation. (a–c) Immunolabelling of S2 cells transfected with *pPac-gal4* and *pUAS-RFP* (Control, (a), or *pPac-gal4*, *pUAS-RFP* and *pUAS- hop*[Tum-l] (*hop*[Tum-l] *GOF*, (b), or *pPac-gal4*, *pUAS-RFP*, *pUAS- hop*[Tum-l] and *pPac-gcm* (*hop*[Tum-l] *GOF, gcm GOF*, (c) and labelled for DAPI (blue), RFP (red) and Stat92E (gray). (a,b,c) show the merge of the three channels, (a',b',c') show RFP alone, (a'',b'',c'') show Stat92E alone and (a''',b''',c''') show DAPI. Arrowheads indicate transfected S2 cells. Scale bar: 20 μm. The percentage of cells presenting nuclear Stat92E is displayed in *Figure 1—figure supplement 4f*. (d) Jak/Stat pathway: inhibitors of the pathway that are regulated by Gcm are in red. (e,f) Relative expression levels of Jak/Stat inhibitors in S2 cells transfected with a *pPac-gcm* expression plasmid (three independent assays) (e) and in embryo *gcm> gcm KD* (f). In both cases, the data are relative to the levels in controls as described in *Figure 1f*. (g) Penetrance of melanotic tumors (n > 50). (h) Relative expression levels of Jak/Stat downstream targets (*apt* (*Starz-Gaiano et al., 2008*), *CG1572* (*Bina et al., 2010*), *CG13559* (*Bina et al., 2010*), *Galphaf* (*Gα73b*) (*Bina et al., 2010*; *Bausek and Zeidler, 2014*) and *slbo* [*Silver and Montell, 2001*]) in hemocytes from stage 16 embryo *srp(hemo)> gcm GOF*. The data are relative to the levels in controls as described in *Figure 1f*.

DOI: https://doi.org/10.7554/eLife.34890.008

To assess the impact of Gcm on the Jak/Stat pathway, we analyzed the levels of expression of down-stream targets of Stat92E upon *gcm* GOF in hemocytes from stage 16 embryos. Several targets of Stat92E were previously described in S2 cells and in hemocytes: *apt* (*Starz-Gaiano et al., 2008*), *CG1572* (*Bina et al., 2010*), *CG13559* (*Bina et al., 2010*), *Galphaf* (*Gα73b*) (*Bina et al., 2010*; *Bausek and Zeidler, 2014*) and *slbo* (*Silver and Montell, 2001*). The over-expression of Gcm significantly reduces the levels of expression of these genes (*Figure 2h*), suggesting that Gcm inhibits the Jak/Stat pathway upon inducing the expression of the inhibitors Ptp61F, Socs36E and Socs44A.

Finally, and most strikingly, Gcm affects the Jak/Stat-mediated lymph gland phenotype (*Figure 3*). The penetrance of lymph glands histolyzed at wandering L3 and the presence of lamellocytes (L4 marker in green) observed in $hop^{Tum-l}$/+larvae are rescued in $hop^{Tum-l}$/+larvae in which Gcm is over-expressed ($hop^{Tum-l}$/+;*gcm GOF*), while they are enhanced in $hop^{Tum-l}$/+;*gcm KD* larvae. Because the penetrance of the $hop^{Tum-l}$/+phenotype is already high in Wandering L3 (~112 hr after egg laying (hAEL)), we hypothesized that down-regulating Gcm might induce a stronger defect at an earlier stage. The difference in lymph gland histolysis between $hop^{Tum-l}$/+and in $hop^{Tum-l}$/+;*gcm KD* larvae is indeed much more significant in Late L3 larvae (~96 hAEL): 46.7% $hop^{Tum-l}$/+larvae and 80.0% $hop^{Tum-l}$/+;*gcm KD* larvae display lymph gland histolysis (*Figure 3g–k*). Thus, silencing Gcm accelerates the lymph gland defects, highlighting the interaction between the two hematopoietic waves and revealing the embryonic transcription factor Gcm as an important player in the process.

In sum, Gcm subverts the inflammatory response due to constitutive activation of Jak/Stat by inducing the expression of inhibitors of this pathway.

## Gcm regulates signaling from the embryonic to the larval hematopoietic wave

We then asked how the embryonic hemocytes signal to the lymph gland. Prime candidates are the pro-inflammatory cytokines of the Upd family since their expression is induced in cells of both hematopoietic waves by wasp infestation (11-fold for Upd2 and 38-fold for Upd3 [*Yang et al., 2015*]) or by septic injury (*Yang et al., 2015*; *Agaisse et al., 2003*) and their mutations prevent the encapsulation of the wasp egg by the fly hemocytes (*Makki et al., 2010*).

$hop^{Tum-l}$ animals that are heterozygous for *upd2* and *upd3* display a reduced penetrance of the melanotic tumor phenotype (*Figure 4—figure supplement 1a*). Moreover, specific Upd2 or Upd3 over-expression in the embryonic hemocytes (*gcm> upd2 GOF* or *upd3 GOF*) is sufficient to induce lymph gland precocious histolysis as well as melanotic tumor formation (*Figure 4a,d*). Thus, cytokine expression solely in the embryonic hemocytes is sufficient to trigger an inflammatory response. Accordingly, conditional down-regulation of *upd2* or *upd3* (*gcm> upd2* or *upd3 KD*) in $hop^{Tum-l}$ rescues the lymph gland phenotypes induced by the $hop^{Tum-l}$ mutation (precocious histolysis and presence of lamellocytes, *Figure 4a–c*).

We next asked whether the inhibitory role of Gcm on the Jak/Stat pathway involves Upd2 and/or Upd3. Gcm inhibits the expression of *upd2* and *upd3*, as their transcript levels increase upon silencing Gcm in hemocytes (*Figure 4e–f*). In addition, transfecting a Gcm expression vector lowers the expression of the two cytokines in S2 cells (*Figure 4g*). Most importantly, Upd2 and Upd3 are epistatic to Gcm in vivo since down-regulating their expression in $hop^{Tum-l}$ and even in $hop^{Tum-l}$/+; *gcm> gcm KD* animals almost abolishes the formation of tumors (*Figure 4h*). Thus, Gcm suppresses the production of pro-inflammatory cytokines in the embryonic hemocytes.

Of note, the tumor phenotype is induced by Upd2 or Upd3 over-expression but not by Gcm silencing, suggesting that threshold levels of the inflammatory pathway may be required for the melanotic tumors to form. Accordingly, the number of circulating hemocytes is higher in larvae over-expressing Upd2 or Upd3 compared to those observed in *gcm KD* animals (*Figure 4—figure supplement 1e*), and so are the levels of the cytokine transcripts (*Figure 4e–f* in the larvae; *Figure 4—figure supplement 1b,b'* in the embryos). Finally, silencing Gcm in animals that over-express either cytokine has a moderate effect on the number of circulating hemocytes, further supporting the view that Gcm does not act as an anti-proliferative factor (*Figure 4—figure supplement 1e*).

Next, we asked how Gcm, Upd2 and Upd3 are connected. Since Upds are the ligands of the Jak/Stat pathway, which is inhibited by Gcm, we hypothesized that their expression also depends on Jak/Stat signaling. Transfecting S2 cells with the Hop$^{Tum-l}$ expression vector strongly induces the expression of *upd2* and *upd3* (*Figure 4g*) and these two loci contain STAT binding sites (*Figure 4—*

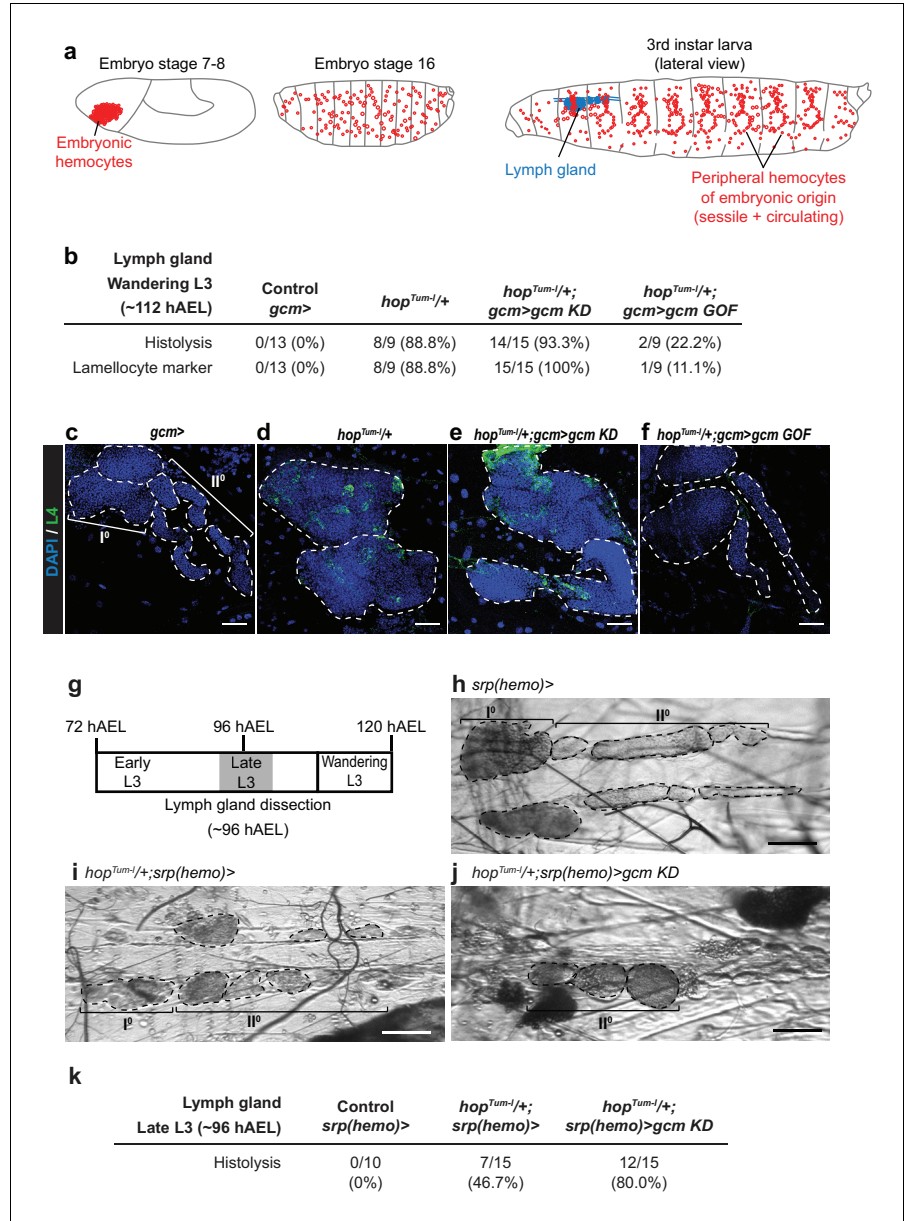

**Figure 3.** Embryonic hemocytes signal to the lymph gland. (**a**) Embryonic hemocytes (red) in early and late embryos as well as in a third instar larva. The lymph gland (blue) histolyzes at the larva to pupa transition. (**b**) Number and percentage of lymph glands showing histolysis and lamellocyte labelling (L4 marker, green) at wandering L3 stage, DAPI is in blue. (**c–f**) Lymph glands are indicated by hatched lines. (**c**) Control lymph gland (*gcm>* ): I° and II° indicate primary and secondary lobes, respectively. (**d,e**) show hypertrophic glands, lack of lobes and L4 expression. (**f**) rescue of the phenotype. (**g–j**) Bright-field images of lymph glands of the indicated genotypes (**h–j**) from Late L3 larva (~96 hAEL) as indicated in (**g**). I° and II° indicate primary and secondary lobes, respectively. Note that the lymph glands are partially histolyzed in $hop^{Tum-l}/+;srp(hemo)>$ (**i**) and $hop^{Tum-l}/+;srp(hemo)> gcm$ KD (**j**) animals. (**k**) Number and percentage of lymph glands showing histolysis at Late L3 stage (~96 hAEL).

DOI: https://doi.org/10.7554/eLife.34890.009

figure supplement 1c,d), calling for a direct role of the Jak/Stat pathway in inducing the expression of these two pro-inflammatory cytokines.

To summarize, Gcm suppresses the Jak/Stat pathway in the primitive wave, thus reducing expression of cytokines Upd2 and Upd3 that can act non-autonomously and signal to the lymph gland.

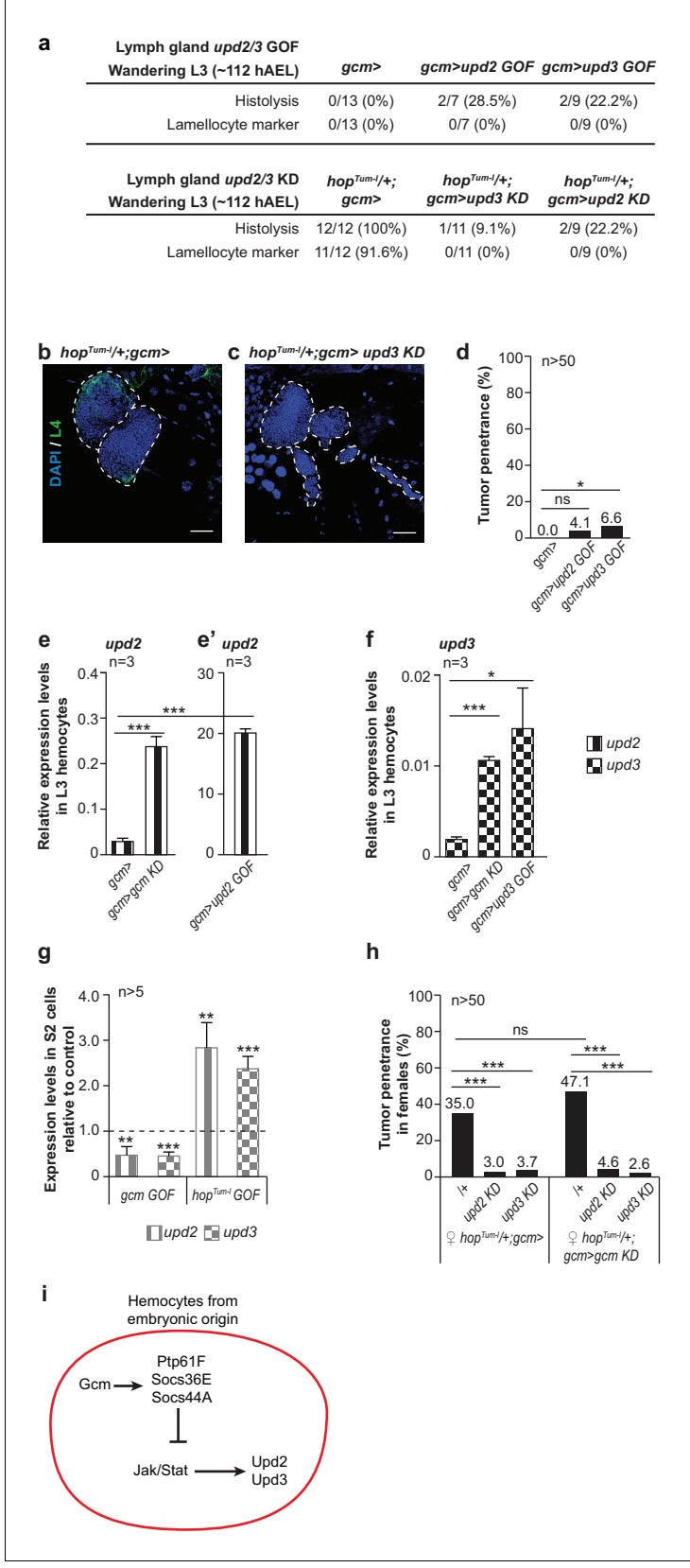

**Figure 4.** Embryonic hemocytes signal through Upd2 and Upd3. (a) Lymph gland phenotypes. (b,c) Lymph gland immunolabelling as in *Figure 3c–f*. (d) Tumor penetrance. (e–f) Relative expression levels of *upd2* and *upd3* in *Figure 4 continued on next page*

*Figure 4 continued*
hemocytes from wandering L3 larvae of the mentioned genotypes. *upd2* and *upd3* expression levels increase in *gcmgcm KD* (e), first two columns from the left in e) and even further in *gcm> upd2/3 GOF* third instar larval hemocytes (e'), column three in (f). Note the different scale between (e) and (e'). (g) *upd2* and *upd3* expression levels in S2 cells upon transfection with *pPac-gcm* or *pUAS-hop*[Tum-l] expression vectors, compared to control levels shown by the dashed line (transfection of an empty expression vector) (n > 5). (h) Tumor penetrance in third instar female larvae. (i) Schematic representation of Gcm contribution to Upd2 and Upd3 production from embryonic hemocytes.
DOI: https://doi.org/10.7554/eLife.34890.010
The following figure supplement is available for figure 4:

**Figure supplement 1.** Interaction between Jak/Stat pathway, Gcm and Upd2/Upd3 cytokines.
DOI: https://doi.org/10.7554/eLife.34890.011

## Jak/Stat activation in the embryonic hemocytes triggers melanotic tumor and lymph gland phenotypes

Based on the above results, we hypothesized that the constitutive activation of the Jak/Stat pathway within the first hematopoietic wave is sufficient to trigger tumor formation/lymph gland defects. Indeed, larvae carrying the *UAS-hop*[Tum-l] transgene and drivers specific to the embryonic hemocytes display tumors, abnormal hematopoiesis and precocious lymph gland histolysis (*Figure 5a–c* third column, *Figure 5—figure supplement 1a–f*). Moreover, silencing the Gcm target and Jak/Stat inhibitor *Ptp61F* in the embryonic hemocytes also triggers tumor formation (3.1% of tumors are detected in *gcm> Ptp61F KD*, n = 128 compared to wild type 0%, n = 182, Khi$^2$ p=0.0164).

Following this, we assessed whether activating the Jak/Stat pathway systemically or only in the embryonic hemocytes (which we will refer to as conditional activation, for the sake of simplicity) produce comparable responses. The penetrance of the tumors is similar (*Figure 5a* third and fourth columns), however, the overall phenotype is stronger in larvae that express *hop*[Tum-l] systemically than in larvae that express *hop*[Tum-l] conditionally: the tumors are more numerous and larger, moreover, 88.8% of the lymph glands are histolyzed and contain lamellocytes (vs. 28% in *UAS-hop*[Tum-l] lymph glands, which contain no lamellocytes) (*Figure 5a,b* third and fourth columns).

To further characterize the role of Jak/Stat activation during the first wave, we analyzed both resident and circulating hemocyte populations. Their total numbers are significantly higher upon systemic Jak/Stat activation (7214 hemocytes in systemic *hop*[Tum-l], 5689 hemocytes in conditional *hop*[Tum-l], p=0.0002, *Figure 5c* third and fourth columns) and the number of dividing cells tends to be higher as well (14.8% vs. 5.1% upon conditional activation, p=0.178, *Figure 5—figure supplement 1c*). Finally, we analyzed the resident and the circulating populations separately, as the inflammatory response is known to trigger hemocyte mobilization from the sessile compartment (*Márkus et al., 2009*; *Zettervall et al., 2004*). Conditional and systemic activation of the Jak/Stat pathway induce lamellocyte markers in hemocytes in both compartments (*Figure 5—figure supplement 1d–f*). However, with the systemic mutation, we observe a depletion of hemocytes from the resident compartment and an increase of the hemocytes in circulation (*Figure 5c* third and fourth column), which indicates abnormal mobilization (*Márkus et al., 2009*; *Zettervall et al., 2004*). This does not occur upon Jak/Stat conditional activation in the embryonic hemocytes (*Figure 5c* third column). In addition, the percentage of resident lamellocytes and the rate of proliferation tend to be higher upon systemic activation (*Figure 5—figure supplement 1c,f*).

In sum, systemic Jak/Stat activation affects the resident compartment more than conditional activation (fewer cells, mostly lamellocytes), likely due to the contribution of the Jak/Stat pathway in additional tissues. Jak/Stat activation in the somatic muscles, which is known to control the inflammatory response upon wasp infestation (*Yang et al., 2015*), does not seem to account for the different phenotypes observed in the sessile compartment as conditional Hop[Tum-l] expression also activates the Jak/Stat pathway in the somatic muscles, as does systemic activation (measured by the *10xStat92E-GFP* reporter, *Figure 5—figure supplement 1g–i*). In line with this, conditional Hop[Tum-l] expression triggers increased levels of the protein accompanied by discrete accumulation in the muscle nuclei (*Figure 6a–b', d*). As predicted, these animals display high levels of expression of the cytokines Upd2/3 in the hemocytes (*Figure 5—figure supplement 1j,k*) (*Yang et al., 2015*).

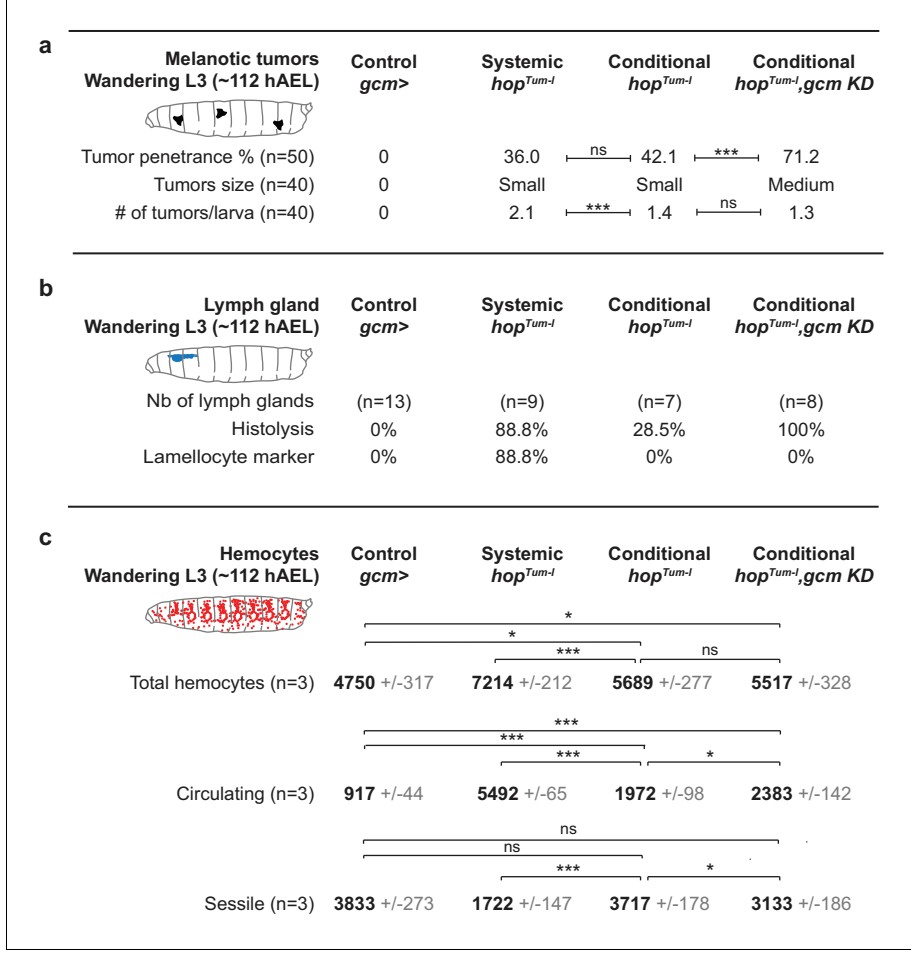

**Figure 5.** Inflammatory response upon systemic and conditional Jak/Stat activation. (a) Tumor penetrance, phenotype expressivity assessed as number of tumors/larva and tumor size of control animals (*gcm>* ), systemic *hop^Tum-l*, conditional *hop^Tum-l* (*gcm> UAS-hop^Tum-l*) and conditional *hop^Tum-l,gcm KD* (*gcm(hemo)> gcm KD,UAS-hop^Tum-l*). The distribution of the tumor size is detailed in *Figure 5—figure supplement 1b*. (b) Precocious lymph gland histolysis and lamellocyte labelling, note the 100% histolysis in conditional *hop^Tum-l,gcm KD*. (c) Total number of hemocytes (circulating +sessile), number of circulating hemocytes and number of sessile hemocytes in systemic and conditional *hop^Tum-l* mutations as compared to controls. n = 3, the numbers in gray indicate s.e.m.
DOI: https://doi.org/10.7554/eLife.34890.012

The following figure supplement is available for figure 5:

**Figure supplement 1.** Phenotypes induced by conditional activation of the Jak/Stat pathway in the embryonic hemocytes.
DOI: https://doi.org/10.7554/eLife.34890.013

## Gcm inhibits the melanotic tumors induced by Hop^Tum-l expression in embryonic hemocytes

Since Jak/Stat activation in the embryonic hemocytes triggers defects in the immune system, the phenotypes, we asked whether silencing Gcm aggravates those defects.

Similar to what observed for *hop^Tum-l*; *gcm KD* animals (*Figures 1* and *3*), reducing Gcm expression in *gcm> hop^Tum-l* animals enhances the phenotype induced by conditional Jak/Stat activation, making it more similar to that triggered by systemic Jak/Stat activation (*Figure 5* fourth column, *Figure 5—figure supplement 1b–f*). The tumor penetrance and expressivity increase and the lymph glands are always precociously histolyzed in conditional *hop^Tum-l;gcm KD* animals (*Figure 5b* fourth column). These phenotypes are not associated with an increase in hemocyte proliferation but with

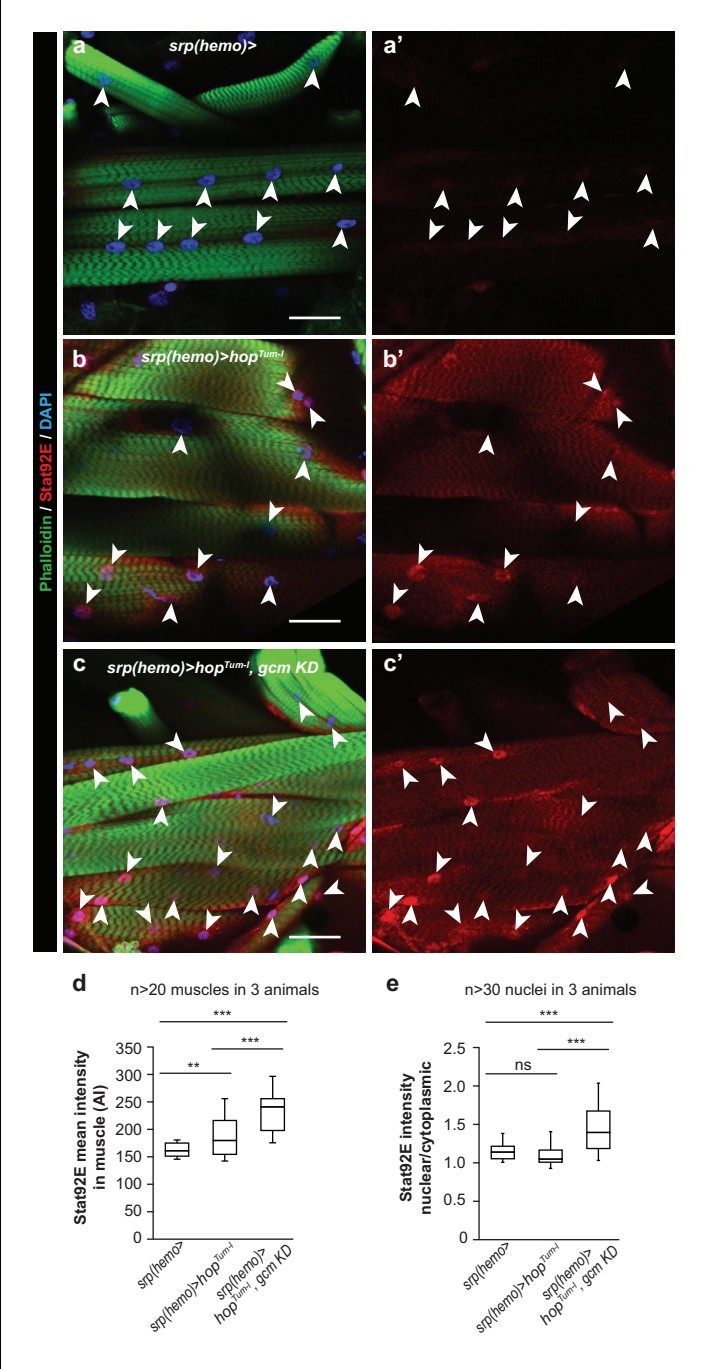

**Figure 6.** Jak/Stat activation in the embryonic hemocytes activates the Jak/Stat pathway in Wandering L3 muscles. (a–c) Muscle immunolabelling in Wandering larvae of the indicated genotypes. The Phalloidin labelling (in green) highlights the striated muscle fibers rich in actin filament, Stat92E is in red and DAPI in blue. (a,b,c) show the merge of the three channels, (a',b',c') show Stat92E alone. Scale bar: 50 μm. Arrowheads indicate nuclei in the muscles. (d,e) Box-plots representing the distribution of Stat92E intensity (d) and the ratio nuclear/cytoplasmic Stat92E (e) in Wandering L3 muscles of the indicated genotypes. Stat92E intensity was quantified in more than 20 muscles in three animals and the ratio nuclear/cytoplasmic Stat92E was measured in more than 30 nuclei in three animals for each genotype. From bottom to top, the boxes represent the 5% percentile, the first quartile, the median, the third quartile and the 95% percentile. P-values were measured using student test after variance analysis.

DOI: https://doi.org/10.7554/eLife.34890.014

enhanced mobilization from the resident compartment (+411 hemocytes in circulation) (*Figure 5c* fourth column). This latter feature is in line with the larger size of the tumors and with the finding that *gcm KD* hemocytes show lower expression of *eater* (*Figure 1f*), a gene required for the attachment of hemocytes to the sessile compartment (*Anderl et al., 2016*). Thus, knocking down *gcm* affects hemocyte mobilization as does systemic Jak/Stat activation. Finally, combining the conditional activation of the Jak/Stat pathway and *gcm KD* further enhances the levels of Stat92E in the muscles and its nuclear localization (*Figure 6b–e*),

In sum, Jak/Stat activation in the first hematopoietic wave triggers an inflammatory response and this response is enhanced in *gcm KD* animals.

## Gcm controls the inflammatory response induced by wasp infestation

We next asked whether Gcm modulates the response to infestation by the wasp *L. boulardi*, which constitutes a more physiological challenge as compared to the chronic challenge induced by a genetically mutant background. Since the immune response to wasp infestation is highly dependent to the conditions applied (*Letourneau et al., 2016*), we first established infestation parameters that trigger comparable fly responses (see Materials and methods section). Our experimental design was adjusted empirically to maximize the number of *Drosophila* larvae containing only one wasp egg.

Following this, we assessed the impact of Gcm on the overall immune response induced by wasp infestation by analyzing wasp egg encapsulation in *gcm KD* condition. Similar to what we found with $hop^{Tum-l}$, Gcm has an anti-inflammatory potential. *gcm KD* mutant larvae show a higher rate of wasp egg encapsulation compared to control larvae, hence allowing fewer wasp eggs to develop. Moreover, fewer adult wasps hatch from *gcm KD* mutant larvae (*Figure 7a,b*). Thus, *gcm KD* animals mount an exacerbated response to an acute challenge.

Since Gcm is no longer expressed by the time of infestation (nor is its expression induced by infestation or by Jak/Stat activation, *Figure 8—figure supplement 1*), this developmental transcription factor likely has priming and long-lasting effects on the embryonic hemocytes. To test this hypothesis, we compared the levels of expression of several sets of genes in *gcm* mutant embryonic and larval hemocytes (*Figure 8*). The levels of the plasmatocyte marker *eater* start decreasing in the mutant embryos and are only significantly low in the larval stage. Similarly, the levels of the lamellocyte markers *βint-v* and *PPO3* are only significantly high in the larva. Also, embryonic mutant hemocytes show a decrease in the levels of the *Ptp61F* Jak/Stat inhibitor and direct target of Gcm (*Figure 2d–f*, *Figure 1—figure supplement 4h*) and this effect is lost at the larval stage, in line with the embryo-specific expression of Gcm (*Figure 8*). This decrease seems sufficient to enhance the expression of the cytokines *upd2* and *upd3* in embryonic stage, which persists in the larval stage (*Figure 8*). The initial effects observed in the embryo become more evident in the larva, likely through positive feedback loops. To eliminate the possibility that we underestimate the embryonic

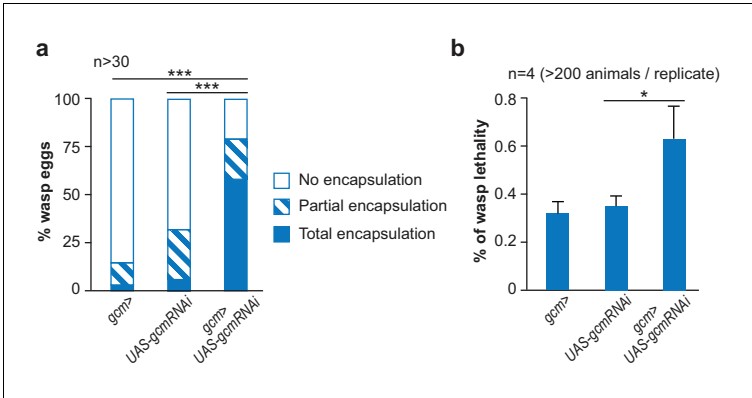

**Figure 7.** The inflammatory response upon wasp infestation is exacerbated in *gcm KD* animals. (**a**) Histogram representing the percentage of total, partial and no wasp egg encapsulation (n > 30). (**b**) Lethality of the parasitic wasp after infestation of *Drosophila* larvae (n = 4) (> 200 animals).
DOI: https://doi.org/10.7554/eLife.34890.015

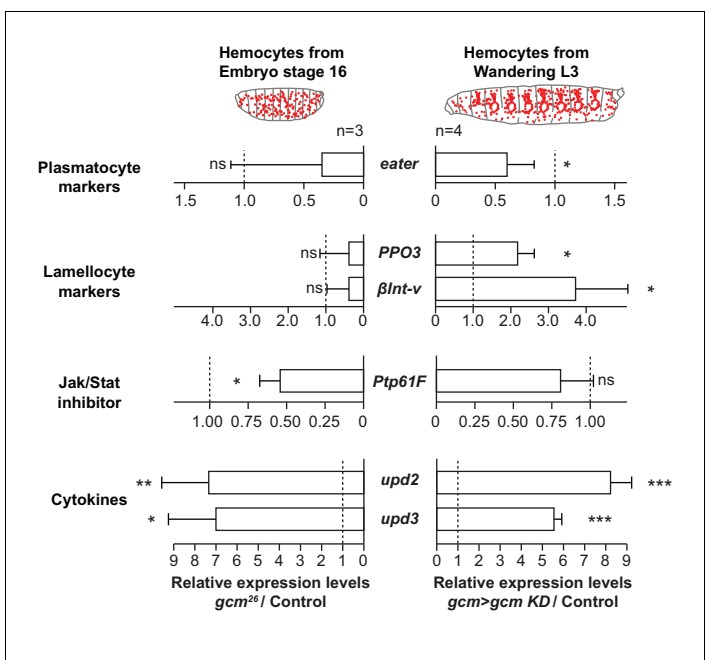

**Figure 8.** Gcm impacts the development and the inflammatory response of hemocytes. Relative expression levels of plasmatocyte and lamellocyte markers, Jak/Stat inhibitor and cytokines in hemocytes from stage 16 mutant embryos (*srp(hemo)> RFP,gcm* [**Cattenoz et al., 2016**]) or from third instar mutant larvae (*gcm> gcm KD*). The levels represented are relative to the levels in controls (*srp(hemo)> RFP* for the embryos and *gcm>* for the larvae) after normalization against housekeeping genes. The data are displayed as described in *Figure 1f*.
DOI: https://doi.org/10.7554/eLife.34890.016

The following figure supplement is available for figure 8:

**Figure supplement 1.** Gcm is not induced in circulating hemocytes nor in lymph glands of 3rd instar larvae upon wasp infestation.
DOI: https://doi.org/10.7554/eLife.34890.017

phenotype due to latency in the RNAi induced defects, we specifically analyzed *gcm* null embryos. Overall, these data suggest that Gcm modulates the level of activation of the Jak/Stat pathway at embryonic stage, which primes hemocytes into a state that is highly sensitive to inflammatory signals in the larval stage.

## Timeline of events triggered by wasp infestation

The above findings strongly suggest that the embryonic hemocytes may represent an early, perhaps the earliest cellular target of wasp infestation. To clarify this issue, we analyzed the timeline of three cellular processes that accompany the response to wasp infestation (*Figure 9a*): the mobilization of the resident hemocytes, the appearance of circulating hemocytes of larval origin, including the appearance of lamellocytes, the formation of melanotic tumors.

At first, we followed the dynamic behavior of the hemocytes after embryogenesis using the *HmlΔRFP* (**Makhijani et al., 2011**) line. In physiological conditions, the hemocytes present in the 1st larval stage do not display a stereotyped arrangement (*Figure 9b*). Hemocytes start homing and localizing sub-epidermally by L2. The so-called sessile pockets become evident by Early L3 and the process continues until Late L3, when the population of resident hemocytes becomes very prominent and stereotyped. Upon wasp infestation, the resident compartments are already disrupted by Early L3 (30 hr after infestation), with many hemocytes being in circulation (*Figure 9c,d*). We quantified the phenotypes and counted in average 7.4 sessile pockets/larva (s.e.m. 0.27) in control and 1.2 pockets (s.e.m. 0.31) in infested animals. Thus, hemocyte mobilization is a very early event.

We next traced the hemocyte lineage within the tumors (*gcm> gtrace* LacZ animals after wasp infestation) and labelled them with the Serpent antibody. In our conditions, the tumors start appearing by Late L3 and seem to mostly contain hemocytes of embryonic origin (Srp positive, LacZ

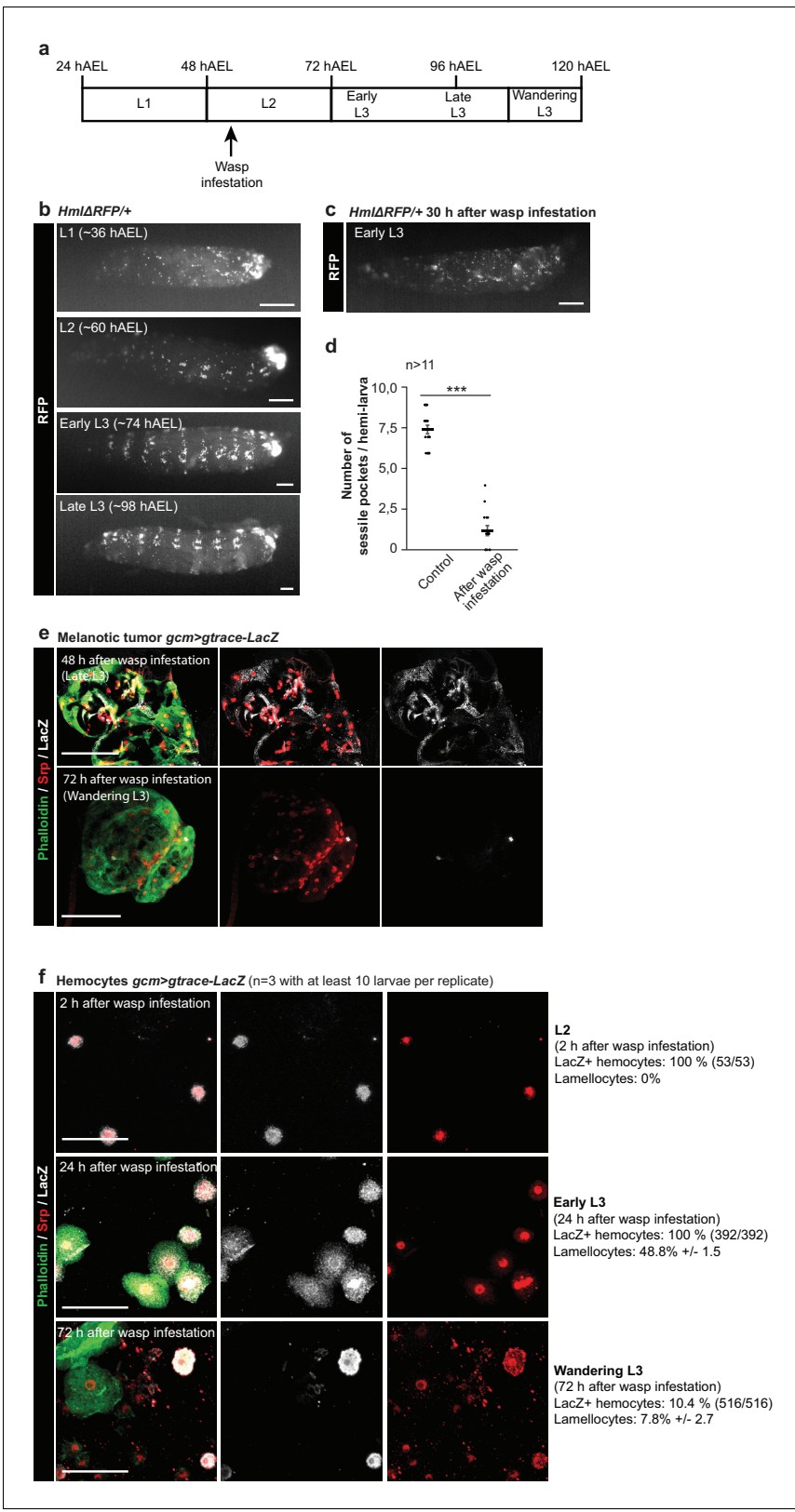

**Figure 9.** Impact of the wasp infestation on the hemocytes. (a) Schematic of the correspondence between the timeline after egg laying (AEL) and the larval developmental stage. The wasp infestation was carried out at the L2 stage. (b–c) RFP signals from *HmlΔRFP/+*larvae at L1, L2, Early L3 and Late L3 stages (b) and Early L3 after wasp infestation (c). The *HmlΔRFP* transgene induces the expression of RFP in hemocytes starting from L1

*Figure 9 continued on next page*

*Figure 9 continued*

(*Makhijani et al., 2011*). The larvae display an accumulation of hemocytes in the sessile pockets from the L2 stage onward. Upon wasp infestation, the sessile hemocytes are released into circulation. Scale bar: 200 µm. (d) Number of sessile pockets per Late L3 larva (one lateral view per larva = hemi larva) in normal conditions and after wasp infestation. Each hemi-larva is represented by a dot, the bar represents the average and the error-bars s.e.m., n > 11, p-value measured using student test after variance analysis. (e) Representative examples of melanotic tumors at different stages. Immunolabelling of tumors from *gcm> gtrace* LacZ larvae, 48 hr and 72 hr after wasp infestation. Phalloidin (green), Srp (red) and LacZ (white). Left panels show the merged channels, mid panels show LacZ and Srp labelling and the right panels show LacZ labelling alone. Scale bar: 100 µm. (f) Immunolabelling of *gcm> gtrace* LacZ larval hemocytes 2 hr, 24 hr and 72 hr after wasp infestation. Phalloidin (green), Srp (red) and LacZ (white). The left panels show the merged channels, the mid panels show LacZ labelling alone and the right panels show Srp labelling alone. Lamellocytes can be recognized by the strong expression of Phalloidin and by their very large size. Scale bar: 50 µm.

DOI: https://doi.org/10.7554/eLife.34890.018

positive) (*Figure 9e*). Tumor melanization (*Figure 8—figure supplement 1d,d'*) hampers proper immunolabelling and quantification of these cells, but clearly their contribution becomes very limited in tumors at later stages (Wandering L3), which are almost completely made of cells from the lymph gland (Srp positive, LacZ negative).

We finally gtraced the embryonic hemocytes (circulating + sessile) and found that they are all of embryonic origin in L2 (2 hr after infestation) (*Figure 9f*). No lamellocyte can be observed by this stage. The hemocytes are all of embryonic origin by Early L3 as well (24 hr after infestation), but by this stage they start to differentiate into lamellocytes (48.8% express a lamellocyte identity). By contrast, most of the circulating hemocytes come from the lymph gland by Wandering L3 (72 hr after infestation), as only 10.4% are of embryonic origin, suggesting that the lymph glands histolyzed. At this stage, fewer lamellocytes are in circulation (7.8%), likely because they are recruited to the tumors that appear in late L3.

Altogether, the developmental and lineage analyses strongly suggest that the sessile hemocytes are mobilized in response to the wasp infestation and start forming the melanotic tumors, to which the hemocytes of the lymph gland contribute at later stages (*Figure 10*).

## Discussion

Immune cells play a key role in physiology and in pathology, therefore understanding how the different hematopoietic waves work is of fundamental importance in basic and medical science. We here show that the embryonic *Drosophila* hemocytes represent major mediators of the immune response by transducing signals that eventually affect the larval hematopoietic organ, the lymph gland. A crucial player of the underlying cellular network is the Gcm transcription factor, which is specifically expressed in the embryonic hemocytes and modulates the response to chronic and acute challenges. This transiently expressed factor does so by, at least in part, regulating the expression of inhibitors of the Jak/Stat pathway. The activation of this pathway in the embryonic hemocytes induces the overproduction of Upd2 and Upd3, two pro-inflammatory cytokines that in turn activate the Jak/Stat pathway in the muscles and the precocious histolysis of the lymph gland. In sum, a molecular cascade specific to the first hematopoietic wave has long-lasting effects that modulate the second wave and the inflammatory response.

### Impact of a developmental cascade on the function of the embryonic hemocytes

Gcm has been defined as a pioneer transcription factor (*Cattenoz and Giangrande, 2015*). Its total loss triggers embryonic lethality associated with the reduction of hemocytes' number and the almost complete lack of glial cells (*Bernardoni et al., 1997*; *Lebestky et al., 2000*; *Alfonso and Jones, 2002*; *Vincent et al., 1996*; *Hosoya et al., 1995*; *Jones et al., 1995*). Using genetic backgrounds that trigger a partial loss of Gcm we could bypass the early defects and reveal its requirement in the immune response (this paper) and in glial migration (*Gupta et al., 2016*), showing that pioneer factors have a much broader spectrum of activities than expected. The late functions (glial migration,

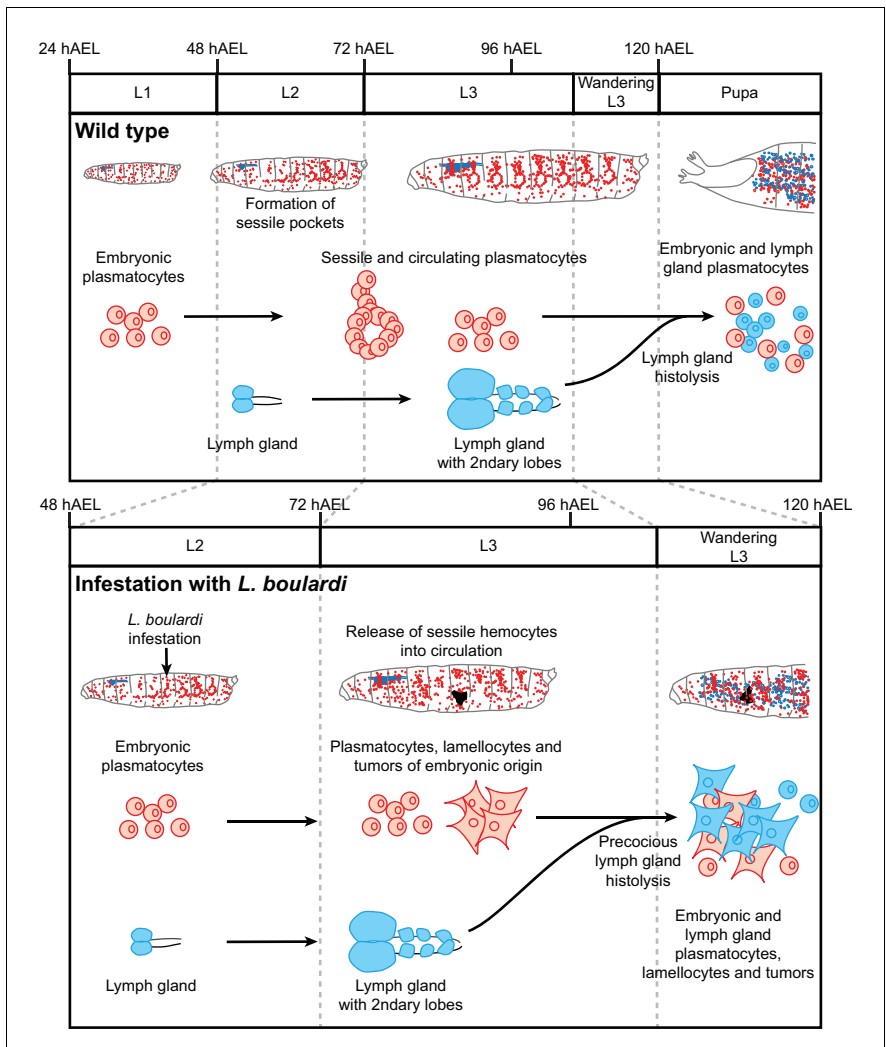

**Figure 10.** Wasp infestation recruits first the embryonic hemocytes and then the LG hemocytes. Timeline summarizing the sequence of events occurring in the immune system of the larva in normal conditions (top panel) and upon wasp infestation (lower panel). In normal conditions, only embryonic hemocytes (in red) are present in the hemolymph until the pupal stage. They start to colonize the sessile pockets by L2 and continue to do so during L3. The lymph gland (in blue) histolyzes at the beginning of pupal stage to release plasmatocytes into circulation. Upon wasp infestation (occurring at L2 stage), the sessile hemocytes are mobilized, embryonic plasmatocytes differentiate into lamellocytes (rhomboid shaped cells) and tumors start forming at Late L3 upon aggregation of the embryonic hemocytes. The lymph gland histolyzes precociously, to release plasmatocytes and lamellocytes that contribute to tumor formation in Wandering L3. Plasmatocytes and lamellocytes originating from the embryo are indicated in red, the ones from the lymph gland in blue. Lamellocytes originating from the embryo are indicated in red, the ones from the lymph gland are in blue.

DOI: https://doi.org/10.7554/eLife.34890.019

hemocyte response to inflammation) require lower levels of Gcm than the early functions (glial and hemocyte differentiation), therefore the potential of this transcription actor depends on the cellular context. Interestingly, the late functions seem associated with a modulatory role. In glial migration, Gcm affects the timely initiation of migration and in the immune response as well, Gcm works as a modulator rather than an instructive factor. Typically, downregulating Gcm does not on its own trigger overt signs of inflammation (hemocyte mobilization, lamellocyte differentiation, melanotic tumors or precociously histolyzed lymph glands).

Gcm allows the proper development of hemocytes that are later able to respond to chronic or acute challenges. Such developmental role fits with the early and transient expression of Gcm in the

hemocyte lineages together with the instability of its transcript and product (*Cattenoz and Giangrande, 2015*). Gcm negatively controls the Jak/Stat pathway upon inducing the expression of the Ptp61F and Socs36E inhibitors in the embryonic hemocytes. We provide several lines of evidence for Jak/Stat pathway inhibition by Gcm to have long-lasting effects. First, over-activating the Jak/Stat pathway specifically in the embryonic hemocytes triggers the formation of melanotic tumors and precocious lymph gland histolysis. Second, silencing Gcm or activating the Jak/Stat pathway in the embryonic hemocytes triggers the overproduction of pro-inflammatory Upd cytokines that act non-autonomously and trigger both tumor formation and precocious lymph gland histolysis. Third, Gcm downregulation triggers the sustained expression of those cytokines in the larval hemocytes, likely through Jak/Stat auto- and cross-regulation. The fact that Gcm expression is not under the control of the immune challenge (Jak/Stat over-activation or wasp infestation) also supports the hypothesis that the observed phenotypes are due to a developmental defect.

Additional factors likely contribute to the long-lasting effects of modulating Gcm expression in the embryo. In addition, the DamID screen identified Gcm direct targets that are required late in hematopoiesis and/or in hemocyte functions, such as phagocytosis, autophagy and defense response to fungi and Gram-negative bacteria (*Cattenoz et al., 2016*). Thus, Gcm may also directly impact late events in the immune system, as is the case in the nervous system where it induces threshold expression levels of a chemoattractant receptor that promotes glial migration (*Gupta et al., 2016*).

Finally, the exacerbated response to acute or chronic challenges due to the low levels of Gcm in the embryonic hemocytes indicate that active repression of the pro-inflammatory Jak/Stat pathway may be required to ensure a fast and efficient response upon challenges.

The finding that altering the expression levels of a developmental factor triggers an inflammatory response that is too strong or too weak, which are both detrimental conditions, constitutes an additional level of control and opens novel perspectives for understanding the molecular mechanisms that modulate the inflammatory response in humans as well. This is particularly exciting in view of recent findings showing that prenatal inflammation and altered development of microglia, the immune cells of the brain, affect the postnatal function of the brain (*Thion and Garel, 2017*).

## Interaction between gcm and the jak/Stat pathway

The role of Gcm is to prevent a state of chronic inflammation induced by the Jak/Stat pathway. Gcm controls the 'inflammatory competence' at least in part by inducing the expression of Ptp61F, Socs36E and Socs44A, which inhibit the Jak/Stat pathway. In so doing, Gcm prevents the production of the Upd2 and Upd3 pro-inflammatory cytokines, whose over-expression is sufficient to induce the inflammatory response (*Figure 11*). Gcm also directly interacts with dPIAS (*Jacques et al., 2009*), which targets Stat92E for degradation via SUMOylation (*Kotaja et al., 2002*) and dPIAS induces melanotic tumors when mutated, due to Stat92E constitutive activation (*Betz et al., 2001*; *Hari et al., 2001*). Thus, Gcm can inhibit the Jak/Stat pathway at least in two ways, by regulating the transcription of the inhibitors, and, in cells in which the Jak/Stat pathway is activated, by binding dPIAS, which in turn inhibits the transcriptional activity of phosphorylated Stat92E.

Despite the multiple levels of regulation, Gcm does not affect all the processes that depend on the Jak/Stat pathway, nor do all the Gcm dependent phenotypes seem to rely on the Jak/Stat pathway: on the one hand, silencing Gcm expression has modest impact on the hemocyte proliferation observed in $hop^{Tum-l}$ animals. On the other hand, the conditional over-activation of the Jak/Stat pathway does not affect hemocyte mobilization, whereas silencing Gcm does, likely through the defective expression of genes required for the attachment of hemocytes to the sessile compartment such as *eater* (*Anderl et al., 2016*). Moreover, Gcm modulates the response to wasp infestation, which involves the Jak/Stat pathway and additional cascades.

How broad is the modulatory role of Gcm in the inflammatory response will await further investigation, but our findings call for a complex cellular network modulating the immune response through multiple organs and tissues including at least the embryonic hemocytes, the muscles and the lymph glands. The future challenge will be to dissect this network and assign specific role to the different tissues and cell types. This will likely include a major player in the humoral response orchestrated by the fat body, which also secretes molecules necessary to mobilize the sessile hemocytes (*Vanha-Aho et al., 2015*).

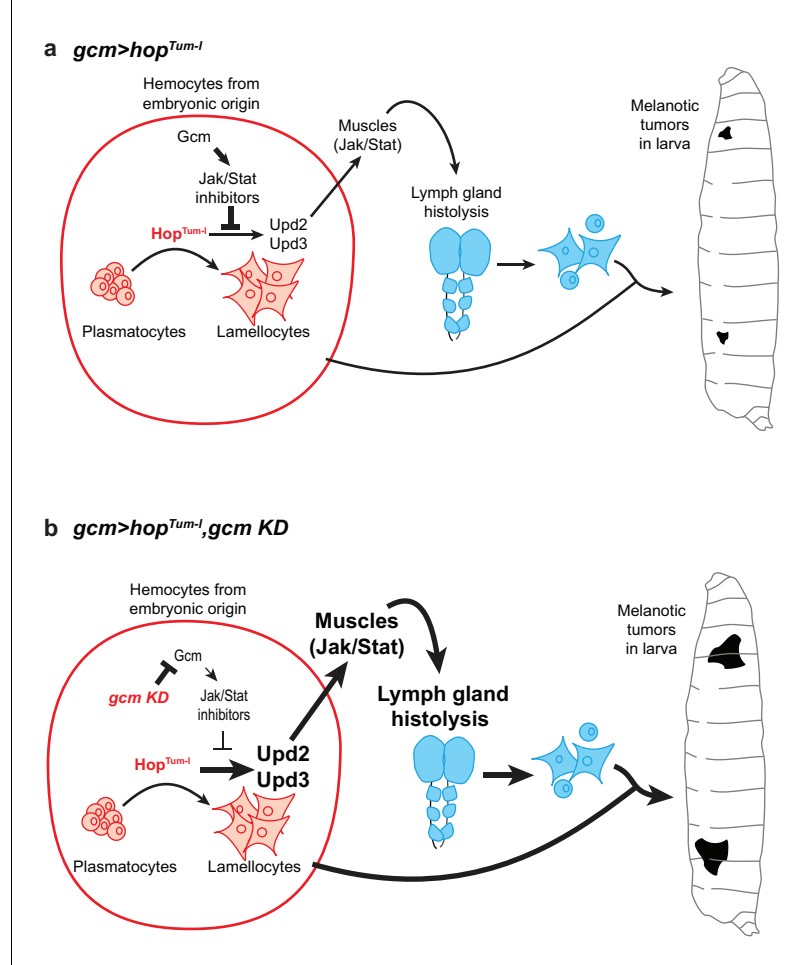

**Figure 11.** Schematic model on the molecular cascade leading to tumor formation upon Jak/Stat activation in the embryonic hemocytes. (**a**) *gcm> hop^{Tum-l}* activates the Jak/Stat pathway exclusively in the hemocytes at embryonic stage. This leads to the production of the Upd2/Upd3 pro-inflammatory cytokines in those cells and to the activation of the Jak/Stat pathway in the larval muscles. This ultimately results in the formation of melanotic tumors and in lymph gland histolysis. When *gcm> hop^{Tum-l}* is combined with *gcm KD* (**b**), the levels of the inhibitors of the Jak/Stat pathway controlled by Gcm (i.e. Ptp61F, Socs36E and Socs44A) decrease in the embryonic hemocytes. This leads to a stronger activation of the Jak/Stat pathway in embryonic hemocytes, which enhances the production of Upd2/Upd3 and the activation of the Jak/Stat pathway in the muscles. As a consequence, *gcm> hop^{Tum-l}*, *gcm KD* animals display stronger phenotypes in terms of lymph gland histolysis and melanotic tumors.

DOI: https://doi.org/10.7554/eLife.34890.020

The Gcm transcription factor provides a new key to investigate the basis of the inflammatory response driven by these complex cellular and molecular interactions. Given the overall evolutionary conservation of the hematopoietic cascades, Gcm orthologous genes may have similar roles. Interestingly, the sea urchin *gcm* orthologue is expressed and required for the differentiation of pigment cells, which function as immune cells (*Calestani et al., 2003*). Focusing on the murine Gcm orthologue will possibly reveal a conserved function in immune/inflammatory responses in higher organisms as well.

## Role of the embryonic hemocytes in the inflammatory response

Previous studies have shown that circulating hemocytes play a central role in the inflammatory responses induced by the fat body (*Pastor-Pareja et al., 2008*; *Vanha-Aho et al., 2015*; *Schmid et al., 2014*) and by the somatic muscles (*Yang et al., 2015*). These circulating hemocytes

comprise hemocytes of embryonic origin and hemocyte released from the lymph gland upon immune challenges. The use of *gcmGal4* and *srp(hemo)Gal4* drivers allowed us to dissect the role and the regulation occurring specifically in the embryonic hemocytes. These cells likely represent the earliest cellular target upon wasp infestation (extending on data from [*Márkus et al., 2009*; *Zettervall et al., 2004*]), in a sequence of events that include the mobilization of resident hemocytes, the formation of tumors and the histolysis of the lymph gland. Together with the finding that *gcm KD* leads to the production of hemocytes that respond much more strongly to inflammatory cues, this demonstrates that an adequate immune response relies on signaling from the embryonic to the larval hematopoietic wave (*Figure 11*).

Embryonic hemocytes may play a primordial role by initiating the inflammatory response, which alerts all the tissues including the larval hematopoietic organs: this was observed by expressing *hop^{Tum-l}* in embryonic hemocytes and noticing the impact on the somatic muscles and the lymph gland. They can also act as transducers of the inflammatory signals between the different organs through the production of the pro-inflammatory cytokines, as knocking down the cytokines in the embryonic hemocytes strongly decreases the systemic inflammation induced by the *hop^{Tum-l}* mutation.

In sum, the present work reveals the homeostatic interactions between hematopoietic waves, which ensure an appropriate inflammatory response to acute or chronic challenges. Given that different hematopoietic waves give rise to immune cells that coexist and are endowed with specific properties in vertebrates, it will be interesting to assess whether similar homeostatic interactions take place in higher organisms.

## Materials and methods

**Key resources table**

| Designation | Source or reference | Identifiers | Additional information |
|---|---|---|---|
| Mouse anti-L4 | Pr. I. Ando | Mouse anti-L4 | (1/30) |
| Rabbit anti-PH3 | Upstate biotechnology #06–570 | Rabbit anti-PH3 | (1/1000) |
| Mouse anti-Hemese | Pr. I. Ando | Mouse anti-Hemese | (1/30) |
| Chicken anti-GFP | abcam #13970 | Chicken anti-GFP | (1/500) |
| Rabbit anti-Serpent | Trebuchet, unpublished | Rabbit anti-Serpent | (1/1000) |
| Rabbit anti-Stat92E | Dr. D. Montell | Rabbit anti-Stat92E | (1/100) |
| Rabbit anti-PPO1 | Dr. WJ. Lee | Rabbit anti-PPO1 | (1/100) |
| Rabbit anti-RFP | abcam #62341 | Rabbit anti-RFP | (1/500) |
| S2 cells | ATCC Ref: CRL-1963 | S2 ATCC | |
| Effectene Transfection Reagent | Qiagen | Effectene Transfection Reagent | |
| WT | Bloomington #5905 | w1118 | |
| hopTum-l | Bloomington #8492 | hopTum-l/FM7c | Point mutation that constitutively activates the Jak/Stat pathway |
| HmlΔRFP | *Makhijani et al., 2011* | HmlΔRFP | |
| UAS-hopTum-l | *Harrison et al., 1995* | UAS-hopTum-l/CyO, twilacZ | Reporter line for hop Tum-l over-expression |
| Gcm> GFP | *Soustelle and Giangrande, 2007* | gcmGal4,UAS-mCD8 GFP/CyO,Tb | Driver specific to embryonic hemocytes and glia, gcm hypomorphic mutation |
| Gcm KD | Bloomington #31519 | UAS-gcmRNAi | dsRNA reporter line for gcm down-regulation |

*Continued on next page*

Continued

| Designation | Source or reference | Identifiers | Additional information |
|---|---|---|---|
| Gcm GOF | *Bernardoni et al., 1997* | UAS-gcmF18A | Reporter line for gcm over-expression |
| Gcm26 | *Vincent et al., 1996* | gcm26/CyOactinGFP | Null gcm mutation |
| Upd2Δ | Bloomington #55727 | upd2Δ | 4.7 kb deletion |
| Upd3Δ | Bloomington #55728 | upd3Δ | Imprecise excision |
| Upd2 KD | Bloomington #33988 | UAS-upd2RNAi | dsRNA reporter line for upd2 down-regulation |
| Upd3 KD | Bloomington #32859 | UAS-upd3RNAi | dsRNA reporter line for upd3 down-regulation |
| Upd2 GOF | *Jiang et al., 2009* | UAS-upd2/CyO | Reporter line for upd2 over-expression |
| Upd3 GOF | *Jiang et al., 2009* | UAS-upd3/CyO | Reporter line for upd3 over-expression |
| Ptp61F KD | Bloomington #32426 | UAS-Ptp61FRNAi | dsRNA reporter line for Ptp61F down-regulation |
| Socs36E KD | Bloomington #35036 | UAS-Socs36ERNAi | dsRNA reporter line for Socs36E down-regulation |
| Socs44A KD | Bloomington #42830 | UAS-Socs44ARNAi | dsRNA reporter line for Socs44A down-regulation |
| Ptp61Fa GOF | *Müller et al., 2005* | UAS-Ptp61Fa/CyO | Reporter line to over-express the cytoplasmic splicing isoform |
| Ptp61Fc GOF | *Müller et al., 2005* | UAS-Ptp61Fc/TM3 | Reporter line to over-express the nuclear splicing isoform |
| Gcm(hemo)> | *Cattenoz et al., 2016* | gcmGal4,UAS-mCD8GFP, repoGal80/CyO | gcm driver not expressed in glia, hypomorphic mutation |
| sn> | *Zanet et al., 2012* | snGal4 | Singed driver, specific to embryonic hemocytes |
| srp(hemo)> | *Brückner et al., 2004* | srp(hemo)Gal4 | Serpent driver specific to embryonic hemocytes |
| dot> | Bloomington #67608 | DotGal4 | Dorothy driver specifically expressed in embryonic and larval lymph gland |
| lz> GFP | Bloomington #6314 | lzGal4,UAS-mCD8GFP | Lozenge driver expressed in crystal cells |
| 10xStat92E-GFP | Bloomington #26198 | 10xStat92E-GFP | Reporter line for STAT activity, 10 Stat92E binding sites driving GFP expression |
| Gtrace | Bloomington #28282 | UAS-FLP;;Ubi-p63E (FRT.STOP)Stinger | This line allows the analysis of lineage-traced expression of Gal4 drivers |
| Ubi> | Bloomington #32551 | UbiGal4 | Expresses GAL4 in all cells |

*Continued on next page*

*Continued*

| Designation | Source or reference | Identifiers | Additional information |
|---|---|---|---|
| RFP | Bloomington #30556 | UAS-RFP | Inserted on the second chromosome |
| Gtrace-LacZ | Bloomington #6355 | P{Act5C> polyA> lacZ.nls1}3 | |
| Fiji | *Schindelin et al., 2012* | Fiji | |
| *pPac-gcm* | *Miller et al., 1998* | *pPac-gcm* | |
| *repoGFP* | *Laneve et al., 2013* | *4.3 kb repo-GFP* | |
| *pUAS-hop^{Tum-l}* | *Harrison et al., 1995* | *pUAS-hop^{Tum-l}* | |
| *pUAS-gcmRNAi* | Vienna Drosophila Resource Center (VDRC) #dna1452 | *pUAS-gcmRNAi* | |

## Fly strains and genetics

Flies were raised on standard media at 25°C. The strains and crosses are detailed in Supplementary Methods.

In all the figures, the larvae were staged as follow: We selected Wandering L3 that never went back to the food, were still very mobile and did not show spiracle eversion (this corresponds to a relatively narrow window around ~112 hAEL). At this stage, 6–8 hr are required for pupariation to occur. All other staging (L1, L2, Early L3, Late L3) were done relative to the time of egg laying: L1:~36 hAEL; L2:~60 hAEL; Early L3:~74 hAEL; Late L3:~98 hAEL. The stages of L2 and Early L3 were further distinguished by observing the morphology of the anterior spiracles as previously described (*Guha and Kornberg, 2005*).

## Penetrance and expressivity of melanotic tumors

Tumor penetrance was determined by assessing the percentage of $3^{rd}$ instar larvae carrying one or more tumors. To assess the expressivity of the phenotype, we classified the tumors into three categories according to their size: Small (S), Medium (M) and Large (L) (*Müller et al., 2005*). A tumor was considered as small when a tiny melanotic mass was documented (*Figure 1—figure supplement 4d*, **left panel**), as medium, when the melanotic mass covered ¼ the distance between the borders of a segment (*Figure 1—figure supplement 4d*, **mid panel**) and as large when the melanotic spot covered ½ the distance between the borders of a segment (*Figure 1—figure supplement 4d*, **right panel**). The expressivity of the melanotic tumor phenotype represents the percentage of small, medium and large tumors counted in each genotype. The p-values were estimated using the chi-squared test for frequency comparisons between two populations.

## Hemocyte counting

Ten third instar larvae were washed in Ringer's solution (pH 7.3–7.4) containing 0.12 g/L of $CaCl_2$, 0.105 g/L KCl, and 2.25 g/L NaCl, then dried, and bled in a 96-well U-shaped microtiter plate containing 50 µL of Schneider medium complemented with 10% Fetal Calf Serum (FCS), 0.5% penicillin, 0.5% streptomycin (PS), and few crystals of N-phenylthiourea ≥98% (PTU) (Sigma-Aldrich P7629) to prevent hemocyte melanization (*Lerner and Fitzpatrick, 1950*). For circulating hemocyte collection, the hemolymph was gently allowed to exit, and the total volume was transferred onto a hemocytometer, where the total number of cells were counted, multiplied by the original volume (50 µL) and the average number of hemocytes per larva was calculated as described in (*Kacsoh and Schlenke, 2012*). For sessile hemocyte collection, the hemolymph containing the circulating hemocytes was

transferred to a first well, while sessile hemocytes were scraped and/or jabbed off the carcass in a second well as described in (*Petraki et al., 2015*) and counted as above. Each counting was carried out at least in triplicates. The p-values were estimated after variance analysis using bilateral student test (see statistics section).

## Hemocyte immunolabelling

Ten third instar larvae were treated as stated above and bled in a 96-well U-shaped microtiter plate containing 200 µL of Schneider medium. Circulating and sessile hemocytes were collected as indicated above and transferred onto a slide using the Cyto-Tek 4325 Centrifuge (Miles Scientific). Samples were then fixed for 10 min in 4% paraformaldehyde/PBS at room temperature (RT), incubated with blocking reagent (Roche) for 1 hr at RT, incubated overnight at 4°C with primary antibodies diluted in blocking reagent, washed three times for 10 min with PTX (PBS, 0.3% triton-x100), incubated for 2 hr with secondary antibodies, washed twice for 10 min with PTX, incubated for 20 min with DAPI to label nuclei (Sigma-Aldrich) (diluted to $10^{-3}$ g/L in blocking reagent), and then mounted in Vectashield (Vector Laboratories). The slides were analyzed by confocal microscopy (see section below on confocal imaging). The following combination of primary antibodies was used to determine the fraction of lamellocytes: rabbit anti-Serpent (1/1000) (Trébuchet, unpublished results) was used to label hemocytes. Serpent is expressed in all hemocyte precursors and is required for the development of plasmatocytes and crystal cells (*Lebestky et al., 2003*). Mouse anti-L4 (1/30) was kindly provided by I. Ando, L4 is an early lamellocyte marker expressed after immune stimulation (*Honti et al., 2010*). The fraction of lamellocytes was determined by counting the number of L4/DAPI positive cells out of the total population of hemocytes present in six confocal fields of vision at 40X magnification and based on Z-series projections. The following combination of primary antibodies was used to determine the fraction of dividing blood cells: rabbit anti-PH3 (1/1000) (Upstate biotechnology #06–570), to assess the mitotic activity, and mouse anti-Hemese (1/30), kindly provided by I. Ando, which recognizes a glycosylated transmembrane protein belonging to the sialophorin protein family and expressed in all larval hemocytes (*Kurucz et al., 2003*). The fraction of dividing cells was determined by counting the number of PH3/Hemese/DAPI positive cells out of the total population of hemocytes, as above. The following combination of primary antibodies was used to determine the fraction of crystal cells: rabbit anti-Serpent (1/1000) and chicken anti-GFP (1/500) (abcam #13970), directed against the membrane GFP signal in *lzGal4,UAS-mCD8GFP* driver expressed in crystal cells. The fraction of crystal cells was determined by counting the number of GFP/Srp/DAPI positive cells out of the total population of hemocytes, as above. Secondary antibodies were: donkey anti-rabbit coupled with Cy3 (1/600) (Jackson #711-165-152), donkey anti-mouse coupled with Cy3 (1/600) (Jackson #715-165-151), goat anti-mouse coupled with FITC (1/400) (Jackson #115-095-166), goat anti-mouse coupled with Alexa Fluor 647 (1/400) (Jackson #115-175-100) and goat anti-rabbit coupled with Alexa Fluor 647 (1/400) (Jackson #711-175-144). Each immunolabelling was carried out on three independent trials. The p-values were estimated after variance analysis using bilateral student test.

## Crystal cell quantification on larval cuticle

Six 3[rd] instar larvae were washed in 1X PBS and heated at 70°C for 10 min in 500 µL of 1X PBS. This procedure leads to the activation of prophenoloxidases (PPOs) within the crystal cells and as a result, these cells appear as black superficial spots on the larval cuticle (*Binggeli et al., 2014*; *Rizki et al., 1980*). 3[rd] instar larval lateral view images were taken under the fluorescent macroscope (Leica, Z16 APO) to cover parts of the dorsal and ventral sides, and superficial crystal cells were counted as described in (*Bretscher et al., 2015*). The p-values were estimated after variance analysis using bilateral student test.

## Cell line

S2 *Drosophila* cells from ATCC Ref: CRL-1963 were used. The identity has been athenticated (STR profiling). No mycoplasm contamination.

## Transfection and qPCR in *Drosophila* S2 cells

Six million (6E6) *Drosophila* S2 cells were plated per well in a 6-well plate with 1.5 mL of Schneider medium +10% FCS+0.5% PS. Transfections were carried out 12 hr after plating, using the Effectene Transfection Reagent (Qiagen) as described in (*Cattenoz et al., 2016*). These transfection assays were used to assess the transactivation potential of (i) Gcm, (ii) Hop$^{Tum-1}$, (iii) the impact of Gcm on Stat92E localization and (iv) to assess the *gcm* RNAi efficiency.

i. To determine the role of Gcm in inducing Ptp61F, Socs36E, Socs44A, Upd2 and Upd3 expression, 2 μg of *pPac-gcm* expression vector (*Miller et al., 1998*) was transfected with 1 μg of *4.3 kb repo-GFP* (*repoGFP*) (*Laneve et al., 2013*): Gcm induces the expression of its target *repo* and drives the expression of GFP, allowing us to recognize and sort the transfected cells (*Laneve et al., 2013*). Co-transfection of 2 μg of *pPac-gal4* driver plasmid and 1 μg of *pUAS-GFP* reporter plasmid was performed as a negative control (*Figures 2e* and *4g*).

ii. To determine the role of *hop$^{Tum-l}$* in inducing Upd2 and Upd3 expression, 0.5 μg of *pPac-gal4* plasmid, 0.5 μg of *pUAS-GFP* and 0.5 μg of *pUAS-hop$^{Tum-l}$* reporters were co-transfected (*Harrison et al., 1995*). Co-transfection of 0.5 μg of *pPac-gal4* driver plasmid and 0.5 μg of *pUAS-GFP*, and 0.5 μg of *pUAS-Empty* was performed as a negative control (*Figure 4g*).

iii. To assess the impact of Gcm on Stat92E localization/expression, S2 cells were transfected as described above with 0.5 μg *pPac-gal4* and 0.5 μg *pUAS-RFP* (Control), or 0.5 μg *pPac-gal4*, 0.5 μg *pUAS-RFP* and 0.5 μg *pUAS- hop$^{Tum-l}$* (*hop$^{Tum-l}$ GOF*), or 0.5 μg *pPac-gal4*, 0.5 μg *pUAS-RFP*, 0.5 μg *pUAS- hop$^{Tum-l}$* and 0.5 μg *pPac-gcm* (*hop$^{Tum-l}$ GOF, gcm GOF*). 48 hr after transfection, the cells were immunolabelled with rabbit anti-Stat92E (1/100, kindly provided by Dr. D. Montell [*Jang et al., 2009*]) and rat anti-RFP (1/500, Chromotek #5F8) as described for the hemocytes (*Figure 2a–c*).

iv. To assess the *gcm* RNAi efficiency, S2 cells were transfected as described above with 0.25 μg *pPac-gal4* driver, 0.25 μg of *pUAS-gcm* expression vector, 0.25 μg of *4.3 kb repo-GFP* (*repoGFP*) (*Laneve et al., 2013*), 0.25 μg of *pUAS-RFP* reporter and 0.25 μg of *pUAS-gcmRNAi* vector (Vienna Drosophila Resource Center (VDRC) #dna1452, used to build the *UAS-gcmRNAi* strain Bloomington #31519). The controls were S2 cells transfected with the same set of plasmids except for *pUAS-gcm* or *pUAS-gcmRNAi* that were replaced by *pUAS-Empty* vector. The levels of GFP and RFP were analyzed 48 hr after transfection using FACSCalibur. The GFP levels were measured in RFP positive cells (*Figure 1—figure supplement 3a*).

For the transfection assays, each combination of plasmids was mixed in 90 μL of EC buffer and 8 μL of enhancer per μg of plasmid followed by 5 min incubation at RT. 25 μL of Effectene was then added and the mix was incubated at RT for 20 min. Then, 500 μL of Schneider medium +10% FCS +0.5% PS was added to the mix followed by spreading it on the cells. For (i) and (ii), plates were then incubated at 25°C for 48 hr followed by sorting on a BD FACSAria, according to GFP or RFP expression to obtain more than 80% of transfected cells in the sample (*Cattenoz et al., 2016*). RNA was then extracted using TRI reagent (Sigma-Aldrich), 1 μg was treated by DNAse1 (RNAse-free) (Thermo Fisher Scientific) and reverse transcribed with Superscript II (Invitrogen). Quantitative PCR (qPCR) assays were performed on a lightcycler LC480 (Roche) with SYBR master (Roche) on the equivalent of 5 ng of reverse transcribed RNA with the primer pairs targeting *Ptp61F*, *Socs36E*, *Socs44A*, *upd2* and *upd3* listed in supplementary methods. Each PCR was carried out in at least three independent replicates. The quantity of each transcript was normalized to the levels of transcripts of two different housekeeping genes, *Glyceraldehyde-3-phosphate-dehydrogenase-1* (*Gapdh1*) and *Actin-5c* (*Act5c*). The p-values were estimated after comparing control to transfected cells using bilateral student test.

## Larval and embryonic hemocyte RNA extraction and qPCR

For embryonic hemocytes, stage 16 embryos carrying *srp(hemo)> RFP* (**see supplementary methods**) were dissociated in PBS using a Dounce homogenizer on ice. The RFP positive cells were then sorted after filtration on a 70 μm filter on a BD FACSAria. At least 10,000 RFP positive cells were used per replicate. For the larval hemocytes, thirty 3$^{rd}$ instar larvae were bled in a 96-well U-shaped microtiter plate containing 200 μL of Schneider medium to collect circulating hemocytes as stated above. Cells were centrifuged at 3000 rpm for 10 min at 4°C. RNA was then extracted using TRI reagent and Quantitative PCR (qPCR) assays were performed on a lightcycler LC480 as stated above

with the primer pairs listed in supplementary methods targeting plasmatocytes markers (*Stofanko et al., 2010*): *crq*, *Hml*, *lectin-24A*, *eater*, *He* and *NimC1*; lamellocyte markers: *Filamin-240 (cher)*, *α-PS5 (ItgaPS5)*, *mys*, *βInt-ν (ltgbn)*, *Tep1*, *Tep4* and *PPO3*; crystal cell markers: *lz*, *hnt (peb)* and *PPO1*; pro-inflammatory cytokines: *upd2* and *upd3*. Each PCR was carried out in at least three independent replicates. The p-values were estimated after comparing control to transfected cells using bilateral student test (see below).

## Lymph gland immunolabelling

Lymph glands from 3$^{rd}$ instar wandering larvae (6 h-8h before pupariation) were dissected in Ringer's solution (pH 7.3–7.4), fixed for 30 min in 4% paraformaldehyde/PBS at 4°C, washed three times for 10 min with PTX, incubated with blocking reagent for 1 hr at 4°C, incubated overnight at 4°C with primary antibodies, washed three times for 10 min with PTX, incubated for 1 hr with secondary antibodies, washed two times for 10 min with PTX, incubated for 20 min with DAPI and then mounted on slides in Vectashield. The slides were analyzed by confocal microscopy (see below). The primary antibody was the mouse anti-L4 (1/30). The secondary antibody was the goat anti-mouse coupled with FITC (1/400) (Jackson #115-095-166). The percentage of precociously histolyzed and lamellocyte expressing lymph glands was assessed. Note that in genotypes carrying the *hop*$^{Tum-l}$ systemic mutation most lymph glands lose their integrity and display only part of the primary and/or secondary lobes because the tissue undergoes precocious histolysis.

## Embryo immunolabelling

*Drosophila* embryos from overnight egg laying at 25°C on apple agar plates were collected, treated and immunolabelled as described in (*Vincent et al., 1996*). They were dechorionated in bleach, rinsed in water then fixed in a solution containing 1 vol. of heptane and 1 vol. of 4% paraformaldehyde in PBS 1x for 25 min. Next, they were devitellinized in methanol and heptane for 1 min followed by treatment with PTX and incubation in blocking reagent for 1 hr at RT. Then, embryos were incubated overnight at 4°C with primary antibodies, washed three times for 10 min with PTX, incubated for 2 hr with secondary antibodies, washed two times for 10 min with PTX, incubated for 20 min with DAPI and then mounted on slides in Vectashield. The slides were analyzed by confocal microscopy (see section below). The following combination of primary antibodies was used to label crystal cells: rabbit anti-PPO1 (1/100) was kindly provided by WJ. Lee. PPOs are essential enzymes in the melanization process, where PPO1 is crystal cell specific marker (*Binggeli et al., 2014*; *Nam et al., 2012*). Chicken anti-GFP (1/500) (abcam #13970) was used to select for right genotype embryos based on *CyOactinGFP* expression. Rabbit anti-RFP (1/500) (abcam #62341) was directed against the RFP signal driven by *lzGal4* driver expressed in crystal cells. Secondary antibodies used were: donkey anti-rabbit coupled with Cy3 (1/600) (Jackson #711-165-152) and donkey anti-chicken coupled with FITC (1/400) (Jackson #703-095-155).

## Jak/Stat expression and activity in larval somatic muscles

Activation of the Jak/Stat pathway was observed in the muscles using the *10xStat92E-GFP* reporter as indicated in (*Yang et al., 2015*). The larvae were frozen and mounted between two slides in water. The images of the larvae were taken at the fluorescent macroscope (Leica, Z16 APO) using 10X magnification and 500 ms of exposure time. The contrast and luminosity of each image were adjusted using Fiji (*Schindelin et al., 2012*); the same correction was applied to all conditions.

In a second approach, Stat92E was directly labelled in striated muscles. Larvae were prepared using the same approach than for the lymph gland described above and labelled with Phalloidin coupled with FITC (1/200), rabbit anti-Stat92E (1/100, kindly provided by Dr. D. Montell [*Jang et al., 2009*]) and DAPI. The images were acquired with Leica SP5 inverted-based confocal microscope using hybrid detectors in photon counting mode and analyzed using the measurement plugin from the software Fiji (*Schindelin et al., 2012*). For total Stat92E quantification, areas covering the muscle on single confocal plans were selected and the absolute intensity/area($\mu m^2$) of Stat92E signal were calculated for at least 20 muscles in three animals per genotype. To calculate the ratio nuclear/cytoplasmic, Stat92E levels were measured in each nucleus individually and coupled with the levels in an area of the same size adjacent to the nucleus. The nuclei were identified using the DAPI signal.

## Wasp survival and encapsulation assays

Wasp parasitization by *L. Boulardi* is commonly used to study the immune response of *Drosophila* (*Yang et al., 2015*; *Small et al., 2012*; *Vanha-Aho et al., 2015*; *Kari et al., 2016*). The wasp lays eggs in the *Drosophila* larva, which induces a strong systemic inflammatory cascade that leads to the differentiation of plasmatocytes into lamellocytes and to the encapsulation of the wasp egg (*Small et al., 2012*). The wasp survival and encapsulation assays were conducted as described in (*Vanha-Aho et al., 2015*; *Kari et al., 2016*) with some modifications.

For wasp survival, 100 first instar *Drosophila* larvae (~24 hAEL) of the indicated genotypes were transferred into a fresh vial at 25°C. At second instar stage (~54 hAEL), 20 couples of *L. boulardi* were added into the vial for infestation for 2 hr, then removed. Following this, the number of wasps hatching from each vial was counted to estimate the % of lethality (1-wasps/*Drosophila* larvae). The sessile pocket observation and the gtrace experiment presented in *Figure 9* were done with this infestation protocol.

For the encapsulation assay, *Drosophila* of the indicated genotypes were allowed to lay eggs for 24 hr at 25°C. The vials containing the embryos were then transferred to 29°C until the second instar stage (48 hr). The *Drosophila* larvae were then exposed to 10 couples of *L. boulardi* for 2 hr at 25°C and after parasitization the vials were incubated at 29°C until the third instar stage. Wandering larvae were dissected to assess the level of melanization of the wasp larvae: total encapsulation (dead wasp larvae completely melanized), partial encapsulation (living larvae, with some melanization), no encapsulation (living larvae, no melanization). Only *Drosophila* larvae containing a single wasp larva were analyzed.

## DamID peaks

The DNA adenine methyltransferase identification (DamID) is an antibody independent method allowing the identification of loci bound by transcription factors (*van Steensel and Henikoff, 2000*; *van Steensel et al., 2001*). Using this approach, the Gcm binding sites in the *Drosophila* genome were recently determined (*Cattenoz et al., 2016*). The peaks indicating Gcm binding onto the *Ptp61F*, *Socs36E* and *Socs44A* loci are represented in *Figure 1—figure supplement 4g–i* using the University of California Santa Cruz (UCSC) Genome Browser (https://genome.ucsc.edu).

## Statistical analysis

The chi-squared test for frequency comparisons between two populations was used to estimate the p-values between percentages of tumors in $3^{rd}$ instar larvae and the expressivity of melanotic tumors in various genotypes tested, where bilateral student test is not applicable. Variance analysis using bilateral student tests for unpaired samples was used to estimate the p-values in hemocyte counting, hemocyte immunolabelling and qPCR assays; in each case, at least three independent trials were performed. In all analyses, 'ns' stands for not significant, for p-value> 0.05; '*' for p-value<0.05; '**' for p-value<0.01; '***' for p-value<0.001. All the P values are in *Figure 1—source data 1*.

## Confocal imaging

Leica SP5 inverted-based microscope equipped with 20, 40 and 63X objectives was used to obtain confocal images. GFP/FITC was excited at 488 nm; the emission filters 498–551 were used to collect the signal. Cy3 was excited at 568 nm; emission filters 648–701 were used to collect the signal, and Cy5 was excited at 633 nm; emission signal was collected at 729–800 nm. A step size between 0.2 and 2 μm was used to collect the Z-series of images, which were then treated with Fiji (*Schindelin et al., 2012*) to obtain fluorescent images using maximum Z-projections. In all images, the intensity of the signals was set to the same threshold in order to compare the different genotypes.

## Acknowledgements

We thank E Sonmez, A Bogomolova, A Pavlidaki and the Imaging Center of the IGBMC for technical assistance, A Bagul for reading the manuscript. We thank I Ando, WJ Lee and D Montell for providing us with antibodies and K Bruckner, M Crozatier, BA Edgar, M Meister, N Perrimon and MP Zeidler for providing fly stocks. In addition, stocks obtained from the Bloomington Drosophila Stock

Center (NIH P40OD018537) as well as antibodies obtained from the Developmental Studies Hybridoma Bank were used in this study. This work was supported by INSERM, CNRS, UDS, Ligue Régionale contre le Cancer, Hôpital de Strasbourg, ARC and ANR grants. P Cattenoz was funded by the ANR and by the ARSEP, W Bazzi by the USIAS and by the FRM (FDT20160435111), V Dasari by COFUND, Y Yuasa by the USIAS and R. Sakr by the ARSEP. This study was supported by the grant ANR-10-LABX-0030-INRT, a French State fund managed by the Agence Nationale de la Recherche under the frame program Investissements d'Avenir ANR-10-IDEX-0002-02.

## Additional information

### Funding

| Funder | Grant reference number | Author |
|---|---|---|
| Institut National de la Santé et de la Recherche Médicale | | Angela Giangrande |
| Centre National de la Recherche Scientifique | | Angela Giangrande |
| Ligue Contre le Cancer | | Angela Giangrande |
| Fondation ARC pour la Recherche sur le Cancer | | Angela Giangrande |
| Fondation pour l'Aide à la Recherche sur la Sclérose en Plaques | Research grant and postdoc fellowship | Angela Giangrande |
| Université de Strasbourg | Graduate student fellowship | Angela Giangrande |
| Agence Nationale de la Recherche | ANR-10-LABX-0030-INRT | Angela Giangrande |

The funders had no role in study design, data collection and interpretation, or the decision to submit the work for publication.

### Author contributions

Wael Bazzi, Conceptualization, Data curation, Formal analysis, Validation, Methodology, Writing—original draft, Conceived and designed the experiments, Performed the experiments, Analyzed the data; Pierre B Cattenoz, Conceptualization, Data curation, Software, Formal analysis, Supervision, Validation, Methodology, Writing—original draft, Conceived and designed the experiments, Performed the experiments, Analyzed the data; Claude Delaporte, Formal analysis, Investigation, Methodology, Performed the experiments; Vasanthi Dasari, Rosy Sakr, Data curation, Formal analysis, Investigation, Methodology, Performed the experiments; Yoshihiro Yuasa, Data curation, Formal analysis, Investigation, Performed the experiments; Angela Giangrande, Conceptualization, Data curation, Formal analysis, Supervision, Funding acquisition, Validation, Investigation, Methodology, Writing—original draft, Project administration, Writing—review and editing, Conceived and designed the experiments, Analyzed the data

### Author ORCIDs

Pierre B Cattenoz (ID) https://orcid.org/0000-0001-5301-1975
Angela Giangrande (ID) http://orcid.org/0000-0001-6278-5120

### Decision letter and Author response

Decision letter https://doi.org/10.7554/eLife.34890.028
Author response https://doi.org/10.7554/eLife.34890.029

## Additional files

### Supplementary files

• Supplementary file 1 details the materials and methods used in the manuscript

DOI: https://doi.org/10.7554/eLife.34890.021

• Transparent reporting form
DOI: https://doi.org/10.7554/eLife.34890.022

**Data availability**
All data are available in the manuscript.

---

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
