## [Decision Letter]

Thank you for submitting your article "Embryonic hematopoiesis modulates the inflammatory response and larval hematopoiesis in *Drosophila*" for consideration by *eLife*. Your article has been reviewed by three peer reviewers, and the evaluation has been overseen by Utpal Banerjee as Reviewing Editor and K VijayRaghavan as the Senior Editor. The following individual involved in the review of your submission has agreed to reveal their identity: Élio Sucena (Reviewer #1).

The reviewers have discussed the reviews with one another and the Reviewing Editor has drafted this decision to help you prepare a revised submission.

Summary:

The manuscript by Dr. Giangrande and coworkers describes the early events during the embryonic wave of hematopoiesis with the larval wave of de novo blood cell formation. The manuscript describes a link in the regulation of inflammation and its threshold of response between events that, in the past, have been described as independent events separated in time and location. The conclusions represent novel advancements to the field of *Drosophila* hematopoiesis. Although the overall perception of the work is positive, several suggestions from the reviewers need to be attended to before the manuscript is reconsidered for publication.

Major changes necessary:

1) There is a persistent gap in linking a causal role played by between Gcm and the series of phenotypes that are attributed to its loss of function. As the authors realize, the number of hemocytes/lamellocytes changes as a consequence of LG integrity or by sessile clusters (SC) recruitment. The number of melanotic tumors is also a consequence of lamellocyte production, which in turn may (or may not) be a consequence of LG histolysis. It is unclear what is/are the phenotype(s) directly caused by GCM? It is important to address this question in order to clearly delineate the mechanism.

2) Related to the above, the broad term "inflammatory response" is neither justified nor necessary in explaining the phenotype. What needs to be made clear, instead, is a careful description of the timeline of events. For example, it is not clear whether histolysis precedes the situations described in Figures 1, 3, 4 and the corresponding text. Similarly, what was the SC status when these hemocyte counts were made? For example, it is unclear in most cases (e.g. Figures 1, 3 and 4, subsection “Gcm regulates signaling from the embryonic to the larval hematopoietic wave”, fourth paragraph) whether there had been histolysis or what was the status of the SCs when haemocyte counts were made. Does gcm directly affect SC recruitment, histolysis or both? Similarly, is the appearance of lamellocytes related to a later event? If the primary cause is histolysis, the term inflammatory response is an exaggeration as being something induced by GCM.

3) The notion that gcm suppresses JAKSTAT pathway, which normally activates the expression of pro-inflammatory cytokines that signal the LG is well supported, but it is important to know what relays the triggering role of gcm at embryogenesis to a process that lasts up to late L3? The persistence of this phenomenon remains unexplored. Is the role of GCM to prevent a state of chronic inflammation by the JAK/SYAY pathway? Could the authors directly test this somehow, perhaps by determining the status of JAKSTAT activation across stages in Hop activated mutants with or without gcm KD?

4) Improve the quality of the figures. This is generally true, but in particular:

a) Figure 1—figure supplement 2, and would be helpful to show a lower magnification picture that shows the whole embryo. Staining is for crystal cells, not plasmatocytes, yet state in the text there is only a slightly reduced number of plasmatocytes. This needs to be shown.

b) What is the purpose of Figure 1F, H? These images don't seem to add much to the paper. Phalloidin is also not a lamellocyte specific marker.

c) The images in Figure 2A, C are out of focus and overexposed. As a consequence, they are uninformative and need to be improved.

d) In Figure 3B and Figure 4A, the quantification of lamellocyte expansion is carried out by scoring the percentage of larvae with L4 staining. This needs to be supplemented with the number of lamellocytes counted in each larva.

e) Figure 1—figure supplement 3H, I: these images add very little to the paper and the quality is rather poor. They should be improved or removed. Figure 8—figure supplement 1A and B need to be improved. Higher magnification images would improve this figure markedly.

5) The use of gcmGAL for "specific" knockdown in the embryonic hemocytes needs to be justified and demonstrated. For example, demonstrate that the drivers do not have even weak expression in the LG. Any expression of the driver in other tissues? Control to show glial expression does not interfere with the phenotype is missing. Also show another hemocyte driver as control to show same effects.

6) Continuing with the above point,

Figure 4A, D: These data are not convincing as numbers analyzed are low. The change in levels or lack thereof needs to be demonstrated by qRTPCR. In general a series of RTPCRs will rule out changes due to LG or glial expression. In Figure 4H: will gcm expression in fat body have the same effect on lymph gland? Or is it residual expression of *gcmGAL4* in LG? How does the hemocyte signal to LG? Is it through fat body or the CNS?

The authors should discuss how the signaling is sustained after expression of gcm in embryonic hemocytes. What about the effect of neural expression? It is not really clear that there is a cascade from only embryonic hemocytes. How do the authors rule out the possibility that a gcm target in nonhematopoietic tissue sustains the inflammation? For example, demonstrate that the effects seen are not because of gcm expression in glia.

7) Figure 2: It would be more relevant to misexpress gcm in vivo in embryonic hemocytes and other tissues, rather than in S2 cells and evaluate pathway inhibitor expression as well as stat activation in the lymph gland. Also, for Figure 2E, F: Without molecular and biochemical analysis it is not possible to comment on GCM transcriptional effect being direct. Also the reduction is significant only for socs. The text should be edited appropriately or additional evidence provided.

---

## [Author Response]

Major changes necessary:1) There is a persistent gap in linking a causal role played by between Gcm and the series of phenotypes that are attributed to its loss of function. As the authors realize, the number of hemocytes/lamellocytes changes as a consequence of LG integrity or by sessile clusters (SC) recruitment. The number of melanotic tumors is also a consequence of lamellocyte production, which in turn may (or may not) be a consequence of LG histolysis. It is unclear what is/are the phenotype(s) directly caused by GCM? It is important to address this question in order to clearly delineate the mechanism.

Point 1, 2 will be dealt with together as they concern related issues.

The reviewers ask what is the timeline of events that involve LG histolysis and what is the role of Gcm. Altogether, our data allow us to propose that Gcm has direct effects on the hemocytes, including a negative impact on lamellocyte differentiation, and that this gene also affects the ability of the hemocytes to communicate with the rest of the body.

To further characterize the direct and the indirect effects of Gcm, we performed additional experiments on the circulating and on the sessile hemocytes as well as on the tumors and on the lymph gland, shown in Figure 2H, 3G-K, 6A-E, 9A-F, Figure 5—figure supplement 1J-K.

1) We describe the timeline of events.

We lineage traced the circulating hemocytes at different times upon wasp infestation and showed that the initial response comes from the embryonic hemocytes, followed by the recruitment of hemocytes of larval origin. We present these data in Figure 9F. By L2 (2 h after infestation), the circulating hemocytes are all of embryonic origin. By early L3 (24 h after infestation), they also are of embryonic origin and half of them express a lamellocyte identity. By wandering L3 (72 h after infestation), most of the circulating hemocytes come from the LG. Overall, fewer lamellocytes are in circulation, likely because they are recruited to the tumors.

2) We analyze the melanotic tumors induced by wasp infestation.

We lineage traced the hemocytes in the tumors and found that those of embryonic origin are the first contributors. We present these data in Figure 9E. By late L3 (48 h after infestation), the tumors are mostly composed of embryonic hemocytes. By wandering L3 (72 h after infestation), most hemocytes in the tumors are of LG origin.

3) We analyze the resident hemocytes during development and upon infestation.

Hemocytes home to the sessile compartment during larval development and are rapidly mobilized upon wasp infestation. We present these data in Figure 9A-D. In physiological conditions, the circulating hemocytes start homing and localize sub-epidermally by L2. This process continues until late L3, when the population of resident hemocytes becomes very prominent and stereotyped. Upon wasp infestation, the resident compartment is already disrupted by early L3 (30 h after infestation), with many hemocytes being in circulation. The early phenotype of the embryonic hemocytes is in line with the lineage tracing and the melanotic tumor data mentioned above.

To summarize, the embryonic hemocytes represent an early, maybe the earliest, cellular target of wasp infestation and are responsible for the initial formation of tumors. The LG is involved at later stages.

We summarize these phenotypes in a new schematic (Figure 10).

4) We analyze the impact of Gcm.

Quantitative analyses of the infestation assays require standardization of several parameters: number of flies/wasps, number of wasp eggs/fly larva, temperature, length of infestation time. For this reason, the role of Gcm in the immune response was assessed in a genetic approach using gcm KD flies in which the JAK/STAT pathway is over-activated. We present these data in Figure 3G-K. See also the response to point 3.

a) Silencing Gcm accelerates LG histolysis. hopTum wandering L3 animals with or without gcm show an equally strong LG phenotype, we hence hypothesized that down-regulating Gcm triggers a precocious defect. Indeed, hopTum LGs histolyze significantly earlier, before the wandering stage, when gcm is down-regulated.

b) Gcm positively controls hemocyte sub-epidermal localization.

i) The proportion of circulating vs. resident hemocytes increases in animals that are gcm> hopTum, gcm KD compared to that found in gcm> hopTum animals. Hence, Gcm helps the correct localization of the embryonic hemocytes (Figure 5C, 3^rd^ and 4^th^ columns). This finding is in line with the reduced expression of Eater in embryonic gcm KD hemocytes (Figure 1F, left panel). Eater is necessary for hemocytes to localize in the sessile compartment (Bretscher, Honti et al., 2015).

ii) Gcm likely affects the hemocytes non-autonomously as well. The levels of edin transcripts increase in the fat body of gcm KD animals (Author response image 1, right panel). Edin codes for a secreted molecule that mobilizes the sessile hemocytes (Vanha-aho, Anderl et al., 2015). Thus, the embryonic hemocytes likely signal (directly or indirectly) to the fat body, although not through the JAK/STAT pathway (see below). We propose this hypothesis in the Discussion.

In sum, the embryonic hemocytes play a major role in the immune response, which also requires the function of and the interaction with other cells and tissues. Gcm affects both the localization of the embryonic hemocytes (directly/indirectly) and the LG (indirectly). The future challenge will be to dissect the complex molecular cascade controlling tissue homeostasis.

2) Related to the above, the broad term "inflammatory response" is neither justified nor necessary in explaining the phenotype. What needs to be made clear, instead, is a careful description of the timeline of events. For example, it is not clear whether histolysis precedes the situations described in Figures 1, 3, 4 and the corresponding text. Similarly, what was the SC status when these hemocyte counts were made? For example, it is unclear in most cases (e.g. Figures 1, 3 and 4, subsection “Gcm regulates signaling from the embryonic to the larval hematopoietic wave”, fourth paragraph) whether there had been histolysis or what was the status of the SCs when haemocyte counts were made. Does gcm directly affect SC recruitment, histolysis or both? Similarly, is the appearance of lamellocytes related to a later event? If the primary cause is histolysis, the term inflammatory response is an exaggeration as being something induced by GCM.

See above.

3) The notion that gcm suppresses JAKSTAT pathway, which normally activates the expression of pro-inflammatory cytokines that signal the LG is well supported, but it is important to know what relays the triggering role of gcm at embryogenesis to a process that lasts up to late L3? The persistence of this phenomenon remains unexplored. Is the role of GCM to prevent a state of chronic inflammation by the JAK/SYAY pathway? Could the authors directly test this somehow, perhaps by determining the status of JAKSTAT activation across stages in Hop activated| mutants with or without gcmKD?

We were also intrigued by the impact of an embryonically expressed gene on a late phenotype. Based on our data, we propose that the embryonic hemocytes send a signal that eventually controls the second hematopoietic wave. In the gcm KD embryonic hemocytes this is, at least in part, mediated by the JAK/STAT pathway that cell-autonomously induces the expression of pro-inflammatory cytokines, which then activate the JAK/STAT pathway in the somatic muscles. Here are the old and the new evidence for this:

1) Gcm induces the expression of JAK/STAT inhibitors in the embryo and in S2 cells (Figure 2E, F).

2) The JAK/STAT pathway can activate the expression of the pro-inflammatory cytokines Upd2/3 in S2 cells (Figure 4G).

3) Gcm inhibits the expression of those cytokines in S2 cells and in the larval hemocytes (Figure 4E-G). These data suggest that Upd2 and Upd3 can relay the triggering role of Gcm, as also confirmed by the LG and tumor phenotypes of animals over-expressing any of the cytokines in the embryonic hemocytes.

At this point, we still missed the demonstration that JAK/STAT activation in the embryonic hemocytes results in high levels of upd2 and upd3 expression in the hemocytes at the larval stage.

4) We filled the gap and showed that the over-expression of the JAK/STAT pathway in the embryonic hemocytes (srp(hemo)> hopTum) leads to increased expression of upd2 and upd3 in those cells at the larval stage (qRT-PCR assays). This validates the hypothesis that JAK/STAT activation cell-autonomously induces the expression of the cytokines and that this is still detectable in L3 hemocytes, well after the pathway has been over-activated. To please the reviewer(s), for these new data we used the srp(hemo) driver. We present these data in Figure 5—figure supplement 1J-K.

5) We extended the data on the non-autonomous role of the JAK/STAT pathway. We analyzed the levels as well as the nuclear localization of the STAT protein in the muscles of srp(hemo)> hopTum larvae. Nuclear STAT is generally associated with high STAT levels and indicates activation of the JAK/STAT pathway. Interestingly, overexpressing hopTum solely in the embryonic hemocytes enhances the overall levels of STAT in the larval muscles but not its nuclear localization. In addition, down-regulating gcm in those animals further enhances the overall levels of STAT and also results in STAT nuclear localization. We present these data in Figure 6. Of note, we see no activation of the JAK/STAT pathway in the fat body of the srp(hemo)hopTum larvae (see Author response image 2). Thus, the JAK/STAT pathway acts directly and indirectly on the immune response (Figure 11).

**Author response image 2. respfig2:** 

6) JAK/STAT activation in the somatic muscles is known to affect the immune response to wasp infestation (Dan Hultmark and collaborators, Yang et al., 2015), a phenotype that involves the lymph gland. We agree with the reviewers that the role of Gcm is to prevent a state of chronic inflammation by the JAK/STAT pathway.

Altogether, our data indicate that the main role of Gcm is to sensitize the embryonic hemocytes and to enhance their response to an immune challenge (hopTum activation, wasp infestation). Gcm inhibits the expression of pro-inflammatory cytokines in the embryonic hemocytes, these cytokines then talk to the muscles, which accelerates LG histolysis.

7) Due to known auto and cross-regulation of the JAK/STAT pathway with other pathways, the levels of upd2/3 remain high even after gcm expression fades away (Figure 4E-F).

We went further and showed that Stat expression is not over-activated in the

(srp(hemo)> hopTum) larval hemocytes (Author response image 3). This may suggest that upd2 and upd3 expression becomes JAK/STAT independent in the 3^rd^instar. These data are not included in the manuscript, which is already quite dense.

**Author response image 3. respfig3:** 

4) Improve the quality of the figures. This is generally true, but in particular:a) Figure 1—figure supplement 2, and would be helpful to show a lower magnification picture that shows the whole embryo. Staining is for crystal cells, not plasmatocytes, yet state in the text there is only a slightly reduced number of plasmatocytes. This needs to be shown.

We complied with the request on the figure. The discrete decrease of hemocyte number in gcm mutant embryos was published by Bernardoni et al. (1997).

b) What is the purpose of Figure 1F, H? These images don't seem to add much to the paper. Phalloidin is also not a lamellocyte specific marker.

We complied with the request of the reviewer(s) and moved this figure to the supplementary material (Figure 1—figure supplement 4A-C’).

We agree that phalloidin is not unique to lamellocytes, but we also used L4, a commonly used lamellocyte marker (Honti, Csordas et al., 2014, Anderl, Vesala et al., 2016). This panel shows that down-regulating gcm induces changes in blood cell specification. The expressivity of the phenotype is low, indicating that Gcm acts as a sensitizing rather than a pioneer (instructive) factor. It is also possible that some cells do not show the full phenotype but are on the go to become lamellocytes, as Hultmark and collaborators (Anderl, Vesala et al., 2016) identified hemocytes with different intermediate phenotypes in challenged conditions.

c) The images in Figure 2A, C are out of focus and overexposed. As a consequence, they are uninformative and need to be improved.

We complied with the suggestions and improved the figures.

d) In Figure 3B and Figure 4A, the quantification of lamellocyte expansion is carried out by scoring the percentage of larvae with L4 staining. This needs to be supplemented with the number of lamellocytes counted in each larva.

We did not count the number of lamellocytes within the lobes of the LG because L4 immunolabelling is cytoplasmic and not nuclear. Individual cells in that case are not easy to identify. Moreover, we cannot compare the number of lamellocytes on histolyzed LG lobes, as this would lead to misinterpretation of the results. To that purpose, we relied on determining the percentage of L4 expressing lobes (LGs). Moreover, we did not count the number of circulating lamellocytes within the larvae where LG analysis was performed due to the fact that those larvae were selected and directly processed for LG dissections. It is not practical to count the total number of lamellocytes and perform LG analysis on the same larvae. On the other hand, we performed total hemocyte counts on other larvae that were not subjected to LG analysis, yet they are of the same genotype (Figure 1D, E; Figure 5—figure supplement 1D-F). Selected larvae were of the same age.

e) Figure 1—figure supplement 3H, I: these images add very little to the paper and the quality is rather poor. They should be improved or removed. Figure 8—figure supplement 1A and B need to be improved. Higher magnification images would improve this figure markedly.

We complied with the requests of the reviewer(s).Figure 1—figure supplement 3H, I has been removed and a higher mag is shown in Figure 8—figure supplement 1A’ and B’.

5) The use of gcmGAL for "specific" knockdown in the embryonic hemocytes needs to be justified and demonstrated. For example, demonstrate that the drivers do not have even weak expression in the LG. Any expression of the driver in other tissues? Control to show glial expression does not interfere with the phenotype is missing. Also show another hemocyte driver as control to show same effects.

We had the same concern as the reviewer, which is why had addressed the issues in three ways:

1) Lack of expression of the gcm Gal4 driver in the lymph gland.

We used a gtrace approach to follow the driver over development and found no expression of the gcm Gal4 driver in the LG, this is shown in Figure 1—figure supplement B, B’. For comparison, note the efficient gtracing of gcm expression in the larval central nervous system (Figure 1—figure supplement 1D). The genotype of these flies is the following: gcm Gal4/UASFLP,UAS-RFP, Ubi-p63E(FRT.STOP)GFP (BDSC #28280).

The lack of lymph gland expression was also published by the laboratory of L Waltzer (AvetRochex, Boyer et al., 2010). Those authors crossed the gcm Gal4 driver with a UAS FLP, Act FRT> CD2> FRT> GAL4 cassette and UAS-GFP reporter to make the gcm Gal4 driven expression of the GFP stable.

2) The observed phenotypes are not due to the glial driven expression of the gcm Gal4 line. We obtained comparable results upon using a gcm Gal4 repo Gal80 line that prevents expression in glial cells. This line is called gcm(hemo)> and the data are shown in Figure 5, Figure 1—figure supplement 4E.

3) We confirmed the gcm Gal4 data by using independent drivers specific for the embryonic hemocytes: srp(hemo) Gal4 and singed Gal4.These data are shown in Figure 1—figure supplement 4E.

6) Continuing with the above point,Figure 4A, D: These data are not convincing as numbers analyzed are low. The change in levels or lack thereof needs to be demonstrated by qRTPCR. In general a series of RTPCRs will rule out changes due to LG or glial expression.

See above for the specificity of the driver. As per the qRT-PCR data, Figure 4E’, F shows the levels of upd2 and 3 transcripts in the larval hemocytes and these data are in line with those in Figure 4A.

In Figure 4H: will gcm expression in fat body have the same effect on lymph gland? Or is it residual expression of gcmGAL4 in LG? How does the hemocyte signal to LG? Is it through fat body or the CNS?

Expressing gcm in the fat body may create artificial phenotypes that could be hardly interpreted. For this reason, we did not perform the experiment.

See above for the potential effects from LG/glia expression of the driver.

See above for the hemocytes signaling to the muscle and to the fat body.

The authors should discuss how the signaling is sustained after expression of gcm in embryonic hemocytes. What about the effect of neural expression? It is not really clear that there is a cascade from only embryonic hemocytes. How do the authors rule out the possibility that a gcm target in nonhematopoietic tissue sustains the inflammation? For example, demonstrate that the effects seen are not because of gcm expression in glia.

See response to point 1.

*7) Figure 2: It would be more relevant to misexpress gcm* in vivo *in embryonic hemocytes and other tissues, rather than in S2 cells and evaluate pathway inhibitor expression as well as stat activation in the lymph gland.*

We complied with this request in two ways:

1) We found that over-expressing gcm in the embryonic hemocytes lowers the expression levels of several targets of the JAK/STAT pathway in those cells. These data are presented in Figure 2H.

Based on the literature, we analyzed the expression of apt, CG1572, CG13559, Galphaf and slbo. All these targets are less expressed in hemocytes that over-express gcm, confirming that Gcm inhibits the JAK/STAT pathway cell autonomously. We did not perform gain of function experiments in other tissues. Cell-specific cofactors necessary for Gcm activity may be absent there, rendering the interpretation of the results uneasy.

**Author response image 4. respfig4:** 

2) Concerning the impact of Gcm on the JAK/STAT pathway in the LG, we clarified this issue in the manuscript.The phenotypes observed upon down-regulating gcm in JAK/STAT over-expressing or infested animals (the formation of tumors, the depletion of the sessile compartments, the histolysis of the lymph gland) may or may not involve JAK/STAT activation in the LG. This is in line with the data from Hultmark and collaborators showing that wasp infestation triggers JAK/STAT activation in the muscle but not in the LG (Yang, Kronhamn et al., 2015). Published work calls for the role of the JAK/STAT pathway in the LG being complex and suggests that wasp infestation actually inhibits that pathway (Letourneau, Lapraz et al., 2016).

Our overall goal is to show that the two hematopoietic waves communicate and that this involves a transcription factor expressed specifically in the embryonic hemocytes.

Also, for Figure 2E, F: Without molecular and biochemical analysis it is not possible to comment on GCM transcriptional effect being direct. Also the reduction is significant only for socs. The text should be edited appropriately or additional evidence provided.

Figure 2E, F show the effect of Gcm transfection in S2 cells and those of gcm down-regulation in embryonic sorted hemocytes.The expression of the three genes is significantly affected by manipulating the expression levels of Gcm in the two assays. This is in line with the similar phenotypes observed in vivo upon KD of any of the three genes (Figure 2G).

We cannot formally exclude an indirect effect, hence we complied with the request and edited the text accordingly. Nevertheless, we previously published that all these genes appear in the Gcm-DAM ID screen aimed at identifying the direct targets of Gcm (Cattenoz, Popkova et al., 2016). Not only do they contain Gcm-DAM ID peaks but also carry Gcm binding sites at their loci. All the other targets identified in the screen were found to be directly regulated by Gcm upon mutagenesis of the Gcm binding sites. Therefore, we are quite confident that is also the case for the JAK/STAT inhibitors.